# Stable isotope evidence for long-term stability of large-scale hydroclimate in the Neogene North American Great Plains

Livia Manser[1], Tyler Kukla[2], and Jeremy K. C. Rugenstein[2]

[1]Department of Earth Sciences, ETH Zürich, Zürich, Switzerland
[2]Department of Geosciences, Colorado State University, Fort Collins, CO, USA

**Correspondence:** Livia Manser (livia.manser@alumni.ethz.ch); Jeremy Rugenstein (jeremy.rugenstein@colostate.edu)

**Abstract.** The Great Plains of North America host a stark climatic gradient, separating the humid and well-watered eastern US from the semi-arid and arid western US, and this gradient shapes the region's water availability, its ecosystems, and its economies. This climatic boundary is largely set by the influence of two competing atmospheric circulation systems that meet over the Great Plains—the wintertime westerlies bring dominantly dry air that gives way to moist, southerly air transported
by the Great Plains Low-Level Jet in the warmer months. Climate model simulations suggest that, as $CO_2$ rises, this low-level jet will strengthen, leading to greater precipitation in the spring, but less in the summer and, thus, no change in mean annual precipitation. Combined with rising temperatures that will increase potential evapotranspiration, semi-arid conditions will shift eastward, with potentially large consequences for the ecosystems and inhabitants of the Great Plains. We examine how hydroclimate in the Great Plains varied in the past in response to warmer global climate by studying the paleoclimate
record within the Ogallala Formation, which underlies nearly the entire Great Plains and provides a spatially resolved record of hydroclimate during the globally warmer late Miocene. We use the stable isotopes of oxygen ($\delta^{18}O$) as preserved in authigenic carbonates hosted within the abundant paleosol and fluvial successions that comprise the Ogallala Formation as a record of past hydroclimate. Today, and coincident with the modern aridity gradient, there is a sharp meteoric water $\delta^{18}O$ gradient with high ($-6$ to $0‰$) $\delta^{18}O$ in the southern Great Plains and low ($-12$ to $-18‰$) $\delta^{18}O$ in the northern Plains. We find that the spatial
pattern of reconstructed late Miocene precipitation $\delta^{18}O$ is indistinguishable from the spatial pattern of modern meteoric water $\delta^{18}O$. We use a recently developed vapor transport model to demonstrate that this $\delta^{18}O$ spatial pattern requires air mass mixing over the Great Plains between dry westerly and moist southerly air masses in the late Miocene—consistent with today. Our results suggest that the spatial extent of these two atmospheric circulation systems have been largely unchanged since the late Miocene and any strengthening of the Great Plains Low-Level Jet in response to warming has been isotopically masked by
proportional increases in westerly moisture delivery. Our results hold implications for the sensitivity of Great Plains climate to changes in global temperature and $CO_2$ and also for our understanding of the processes that drove Ogallala Formation deposition in the late Miocene.

# 1 Introduction

The Great Plains of North America rise slowly in elevation from the wooded lowlands near sea-level west of the Mississippi River to the greater than 4 km high 'purple mountain majesties above the fruited plain' (Bates, 1911) that comprise the North American Cordillera. Though this region contains some of the flattest landscapes in the United States (Fonstad et al., 2007; Dobson and Campbell, 2014), these Plains belie a remarkable climatic setting and geologic history which have conspired to shape the modern-day water resources, ecosystems, and economies of the Plains region. Boreal spring heralds the onset of the Great Plains Low-Level Jet (GPLLJ), a primarily nocturnal, southerly jet responsible for transporting more than 30 percent of the water vapor that enters the continental US every year (Helfand and Schubert, 1995). Interactions between the GPLLJ and mid-latitude storm systems yield some of the largest and most intense convective systems on the planet (Song et al., 2019). These spring and summertime rains nurture the vast grasslands of the Plains, which return this moisture to the atmosphere via transpiration, often seeding additional precipitation on subsequent days and resulting in one of the tightest couplings between land and atmosphere anywhere on Earth (Koster, 2004). On multi-annual timescales, these same interactions—modified by long-term climatic oscillations such as the El Niño Southern Oscillation or the North Atlantic Oscillation—are the proximal cause for the extensive floods and deep droughts that frequent this region (Byerle, 2003), perhaps best exemplified by the 1930s Dust Bowl (Schubert, 2004), an event which reshaped American governance and society and has been labeled the worst environmental catastrophe in US history (Egan, 2006).

Climatically, the Great Plains are characterized by a sharp aridity gradient with a more humid climate to the east and a more arid climate to the west (Fig. 1). This aridity gradient spans the 100th meridian and marks a dramatic change in the long-term patterns of precipitation, vegetation, and, consequently, of agriculture and human development (Powell, 1879, 1890; Webb, 1936; Seager et al., 2018b). A convenient measure of aridity is the aridity index (AI), which is the ratio of precipitation (P) to potential evapotranspiration (PET): conditions are considered arid—or water-limited—if P/PET is less than one. The Great Plains today straddle the transition between the wet, eastern US (AI > 1) and the water-limited western US (AI < 1) (Seager et al., 2018b). Consequently, regions to the east are more densely populated and farms rely on rainfed agricultural practices; to the west, settlement is more limited and agriculture relies extensively on groundwater withdrawals or irrigation diversions.

This groundwater—sourced from the Ogallala Aquifer (black polygon outlined in Fig. 1) and replenished by spring and summertime rains—is hosted in the Ogallala Formation, a nearly continuous formation that underlies the Plains from southwest Texas and southeast New Mexico into the southern part of South Dakota, making it one of the largest and most laterally continuous sedimentary formations in North America. Comprised of sediments shed off the Rockies, the Ogallala Formation has been interpreted as a series of alluvial or telescoping megafans (Seni, 1980; Willett et al., 2018; Korus and Joeckel, 2023a) that prograded towards the east and completely buried the pre-existing, erosional landscape that, in the southern Great Plains, is cut into Triassic and Permian bedrock, and, in the northern Great Plains, lies on the White River or Arikaree Groups. In places, the Ogallala Formation is overlain by alluvial and/or eolian deposits and, particularly in the southern Great Plains, there is frequently a prominent calcium carbonate caprock that separates the Ogallala from overlying sedimentary units (Gustavson and Winkler, 1988). Deposition began in the middle Miocene and ended in the late Miocene or earliest Pliocene. However,

precisely why the Great Plains experienced a prolonged period of deposition, followed by a period of incision that continues to the present-day remains uncertain, with most studies attributing this period of deposition to dynamic topography effects associated with passage of the Farallon Plate beneath the Great Plains (Moucha et al., 2009; Karlstrom et al., 2011; Willett et al., 2018).

During this time, substantial ecological changes occurred, largely yielding the pre-anthropogenic landscape that characterized the Quaternary Great Plains. Though grasslands were likely present before the middle Miocene, they continued expanding throughout the Plains during the cooling following the peak of mid-Miocene warmth (Jacobs et al., 1999; Stromberg, 2005; Strömberg and McInerney, 2011). These predominantly $C_3$ grasslands were progressively replaced by $C_4$ grasslands starting in the latest Miocene and through the Pliocene (Fox and Koch, 2003, 2004). Largely coincident with these changes, large mammal diversity has gradually declined on the Plains from its peak during the middle Miocene to a relative low in the Quaternary (Janis et al., 2000; Fritz et al., 2016). Since cessation of Ogallala deposition, a combination of base-level fall and uplift along the Front Range led to incision of the major Plains rivers through the Ogallala Formation (McMillan et al., 2002; Duller et al., 2012), leaving the former Ogallala landscape abandoned and perched above the major Plains valleys (Willett et al., 2018). Today, much of the Ogallala Formation is visible as a prominent escarpment protected by the indurated nature of its many calcic-rich sediments; dotting much of the length of this escarpment are wind turbines powered, to no small extent, by the exceptionally predictable and windy Great Plains Low-Level Jet.

This combination of climate and geology has helped to promote development and agriculture on the Plains, with Ogallala Aquifer water supplying any deficit due to insufficient rainfall. However, the aquifer remains in many places critically overdrawn; further, anthropogenically driven increases in atmospheric $CO_2$ and associated changes in global climate may affect water availability on the Plains. Indeed, this area appears to be exceptionally sensitive to even small changes in climate and land cover due to the tight coupling between land and atmosphere in this region (Koster, 2004; de Noblet-Ducoudre et al., 2012; Lague et al., 2019). Thus, small changes in precipitation and/or PET may shift the precise location of the "climatological 100[th] meridian" (*i.e.*, the approximate location of the boundary between arid and wet ecosystems) with important consequences for ecosystems and agricultural systems. Further, global climate model (GCM) simulations tend to predict only a small change in summer precipitation, though a large shift in the seasonality of that precipitation, driven by dynamical shifts in the westerly jet and the GPLLJ (Cook et al., 2008; Bukovsky et al., 2017; Zhou et al., 2021a). Combined with rising temperatures, this negligible change in summer precipitation implies that the arid conditions characteristic of the western Great Plains may expand eastward with global warming as increases in PET outpace increases in P over the Plains (Seager et al., 2018b; Overpeck and Udall, 2020). However, models still struggle to properly simulate the GPLLJ and its associated precipitation-bearing convective systems. Changes in the interactions between vegetation, soil moisture, and rainfall at higher atmospheric $CO_2$ remains similarly difficult to model (Bukovsky et al., 2017; Zhou et al., 2021a). Further, paleoclimate data indicates that, in general, warmer periods have actually been wetter and/or greener (Caves et al., 2016; Burls and Fedorov, 2017; Ibarra et al., 2018; Feng et al., 2022), in conflict with many model predictions of drier, future conditions (*e.g.* Scheff, 2018). As a consequence, there exists substantial uncertainty regarding how aridity on the Plains—and the interactions between the GPLLJ, precipitation, and vegetation—will change as atmospheric $CO_2$ rises and global temperature increases.

In this contribution, we take advantage of the remarkable spatial extent of Neogene sediments afforded by the Ogallala Formation to understand how changes in global climate impacted the Plains during periods of higher atmospheric $CO_2$ and warmer global temperatures. Mid-Miocene global temperatures were 7-8 °C warmer than today and atmospheric $CO_2$ levels were around 500 ppm or higher (Herbert et al., 2016; Steinthorsdottir et al., 2021; Hönisch et al., 2023). Since the mid-Miocene, temperature and atmospheric $CO_2$ have gradually declined, establishing the bi-polar glaciation that characterizes Quaternary climate. The Neogene therefore provides an opportunity to answer the question of how aridity has changed on the Plains and whether the "climatological 100[th] meridian" shifted eastward in the past as global climate models suggest for the future. The spatial extent of the Ogallala also permits us to examine the mechanisms by which any shifts in aridity may have occurred, due to, for example shifts in the distribution of precipitation as a result of strengthening or weakening of the GPLLJ in a warmer climate. Further, any shifts in climate may point towards the mechanisms that generated the sediment that formed the Ogallala Formation, permitting distinctions between climatically or tectonically controlled generation of the Ogallala Formation (*i.e.*, Molnar and England, 1990; Zhang et al., 2001).

To answer these questions, we rely on the stable isotopes of authigenic carbonates—a material that is particularly abundant within the fluvial and paleosol facies of the Ogallala Formation and which are thought to record precipitation $\delta^{18}O$. Precipitation oxygen isotopes ($\delta^{18}O_p$) are sensitive to the moisture source and rainout history of airmasses that reach a given location. Because of this sensitivity, spatially-resolved datasets of carbonate oxygen isotopes ($\delta^{18}O_c$) interpreted to track $\delta^{18}O_p$ have been successfully used to constrain how large-scale hydroclimate has changed in response to atmospheric and orographic forcing through time (Fox and Koch, 2004; Mix et al., 2011; McDermott et al., 2011; Kocsis et al., 2014; Caves et al., 2015; Caves Rugenstein and Chamberlain, 2018). The Plains today host a steep $\delta^{18}O$ gradient oriented NE-SW that roughly tracks, but is somewhat oblique to, the aridity gradient at the 100[th] meridian (Figure 1C). Given this concurrence, we hypothesize that the Ogallala authigenic carbonate $\delta^{18}O_c$ record over space and time will reflect changes in the large-scale hydroclimate and aridity over the Great Plains. In the following sections, we explain how $\delta^{18}O$ can be used to track hydroclimate and our approach to sampling and to building a spatially-extensive dataset. We then investigate the climatic drivers behind the modern $\delta^{18}O_p$ gradient over the Plains and compare this to reconstructed maps of $\delta^{18}O_p$ during Ogallala deposition. Lastly, we apply a recently developed reactive transport model to quantitatively assess our observations of $\delta^{18}O$ and how these data relate to overall hydroclimate on the Plains. We find that large-scale atmospheric circulation and the position of the "climatological 100[th] meridian" have been remarkably stable, despite correspondingly large changes in global climate since the late Miocene.

## 2 Background

### 2.1 Modern hydroclimate of the North American Great Plains

Spring and summer precipitation in the Great Plains overwhelmingly originates from the Gulf of Mexico, where the Bermuda High drives southeasterly flow while a high pressure system over the Great Basin effectively blocks Pacific moisture. The combination of these two high pressure systems create the conditions for the Great Plains Low-Level Jet, which carries moisture deep into the interior of the North American continent (Helfand and Schubert, 1995). Though the precise mechanisms

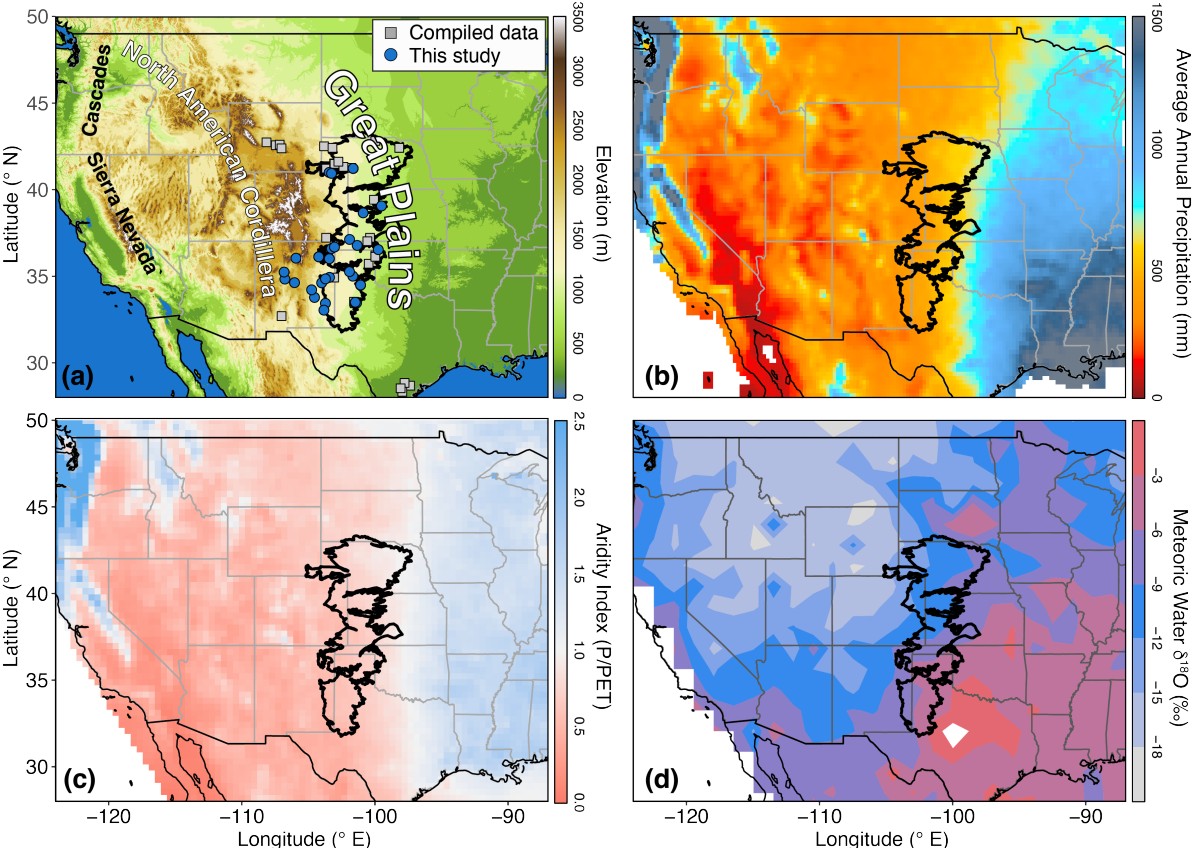

**Figure 1.** (a) Extent of the continuous Ogallala Formation (thick black outline), with new sites presented in this study (squares) and sites compiled from the PATCH Lab (circles) (Kukla et al., 2022a). Shading is elevation (m). (b) Map of average annual precipitation (mm) from the Global Precipitation Climatology Project (Meyer-Christoffer et al., 2011). (c) Aridity index (AI), which is calculated as P/PET. P as in panel (b); PET is from the Global Land Evaporation Amsterdam Model (GLEAM) (Miralles et al., 2011; Martens et al., 2017). (d) Interpolated distribution of $\delta^{18}O$ of modern meteoric waters, derived from rivers/streams, groundwater, tapwater, springwater, and precipitation $\delta^{18}O$ measurements and compiled from the Waterisotopes.org database (Waterisotopes Database, 2019). Thick black outline in all panels is the areal extent of the continuous Ogallala Formation.

that generate the GPLLJ are complex (see Shapiro et al., 2016), critically for our purposes, both the existence of the high topography of the North American Cordillera and the east-west sloping terrain of the Great Plains appear crucial for generating and maintaining a low-level jet over the Plains. For example, in model simulations where Cordillera topography is removed, the GPLLJ is weakened or non-existent (Ting and Wang, 2006; Jiang et al., 2007). During winter, the GPLLJ is inactive and precipitation originates largely from the Pacific Ocean due to storms routed by the mid-latitude westerly jet. These wintertime storms traverse the wide expanse of topography that comprises the North American Cordillera—including ranges such as the Sierra Nevada, Cascades, and Wasatch—which removes moisture from these mid-latitude cyclones. Consequently, precipita-

tion across the Plains typically occurs during interaction with cold Arctic air-masses that can penetrate as far south as the southern Great Plains (Nativ and Riggio, 1990; Brubaker et al., 1994). Thus, the combination of topography with atmospheric circulation generates much of the seasonal precipitation pattern that prevails today over the Great Plains.

The position of the 100th-meridian aridity boundary is shaped by the relative strength of these two circulation systems (Seager et al., 2018b), yielding a sharp humidity gradient that approximately coincides with the boundary between southerly maritime, Gulf of Mexico air and dry continental air (Hoch and Markowski, 2005). On an interannual basis, this boundary can shift depending upon the strength of the mid-latitude westerlies relative to the GPLLJ; with increasing westerly wind strength, for example, the 100th meridian shifts east (Hoch and Markowski, 2005). On longer timescales, winter aridity and weaker

westerlies have been linked to grassland expansion and forest dieback in the Great Plains in the early Miocene (Kukla et al., 2022b). Besides these two large-scale atmospheric systems, several other factors determine the location and orientation of the sharp aridity gradient that characterizes the Great Plains today (Seager et al., 2018b). First, evapotranspiration of water from the land surface to the atmosphere is critical in determining precipitation. For example, return of water to the atmosphere from the land surface (largely through transpiration) may supply up to 40 percent of the precipitation in the Great Plains during

spring and summer (Burde et al., 2006; van der Ent et al., 2010). The importance of the land surface may be further enhanced by the widespread grasslands that populate the Plains landscape. Grasses can much more rapidly modify their stomatal conductance and, hence, total transpiration than can trees and shrubs to take advantage of periodic rainstorms (Ferretti et al., 2003; Hetherington and Woodward, 2003). Such rapid water use leads to higher recycling rates of water from the land surface back to the atmosphere. As a consequence, the spread of grasslands onto the Plains during the Miocene has been hypothesized to have

fundamentally increased the recycling of water between the land surface and the atmosphere (Mix et al., 2013; Chamberlain et al., 2014).

    Model simulations project distinct changes in precipitation and hydroclimate associated with dynamical and thermodynamic responses to warming. GCMs and Regional Climate Models robustly predict that precipitation seasonality will shift from the summer to the spring (Cook et al., 2008). As global temperatures rise, the westerly jet shifts poleward, permitting a stronger

and more northerly GPLLJ, producing more precipitation in the late spring (Bukovsky et al., 2017; Zhou et al., 2021a). In contrast, the continued northward shift of the GPLLJ as summer progresses weakens the jet over the Great Plains, leading to enhanced late summer drying (Zhou et al., 2021a). Despite this shift in the timing of the wet-season, mean annual precipitation is not expected to change (Bukovsky et al., 2017). With a constant mean annual precipitation, the increase in PET owed to rising temperatures will decrease the AI and shift of the "climatological" 100th meridian eastward (Seager et al., 2018a).

**2.2   Precipitation oxygen isotopes reflect this hydroclimatic pattern**

This annual mixing between the GPLLJ and the westerlies results in a steep spatial gradient in precipitation $\delta^{18}O$ mostly due to the differences in topography traversed by each air-mass (Kendall and Coplen, 2001). Mountain ranges tend to increase the net loss of moisture from an airmass, preferentially removing $^{18}O$ and decreasing $\delta^{18}O_p$ (Rozanski et al., 1993; Page Chamberlain and Poage, 2000; Poage and Chamberlain, 2001; Winnick et al., 2014; Kukla et al., 2019). Thus, precipitation derived from

165 the westerlies contains low $\delta^{18}O$ by the time it reaches the Great Plains. In contrast, GPLLJ moisture, which has not traversed

major topographic barriers and is augmented by a high degree of evapotranspiration that replenishes the GPLLJ, is about 10‰ higher than equivalent westerly moisture (Mix et al., 2013; Winnick et al., 2014). The aridity gradient, which depends on the relative contributions of westerly vs GPLLJ moisture, is therefore encoded in the $\delta^{18}O_p$ data which can generally be understood as the precipitation-weighted average of the end-member sources. This precipitation $\delta^{18}O$ signal is captured by authigenic carbonates, but is further modified by both the temperature of carbonate formation and other, potentially spatially variable factors associated with mineral formation, such as differences in precipitation seasonality and evaporation (Breecker et al., 2009; Caves, 2017; Huth et al., 2019; Kelson et al., 2020, 2023).

## 3 Geological setting and sampling approach

The sediments of the Ogallala Formation originate from the Rocky Mountains, and eroded material from the Miocene Rockies was transported by braided, high-energy, ephemeral streams and eolian processes across the Plains (Joeckel et al., 2014; Smith and Platt, 2023; Korus and Joeckel, 2023b). The result was the Ogallala Formation, which spans from South Dakota to southern Texas (black outline in Fig. 1). Though the headwaters of these rivers and fans have since been eroded away, except in southern Wyoming, discontinuous remnants of the Ogallala have been mapped nearly up to their sources in eastern New Mexico (Frye et al., 1982). Consisting of gravel, sand, silt and clay deposits, the Ogallala Formation also contains abundant calcic paleosols and calcic-rich sediments distributed throughout the formation and across the entire N-S extent of the formation (Gustavson, 1996; Joeckel et al., 2014; Smith et al., 2016; Smith and Platt, 2023). The Ogallala Formation in the southern Great Plains unconformably overlies Permian through Cretaceous strata (Gustavson and Holliday, 1999). In the northern Great Plains, the Ogallala Formation is underlain by the late Oligocene to early Miocene Arikaree Group and the White River Group, upper Eocene to Oligocene in age. Chronostratigraphy of the Ogallala Formation is based on fossil vertebrate faunas of Barstovian to Hemphillian North American Land Mammal Ages (NALMAs) and scattered volcanic ash beds and basalt flows (Kitts, 1965; Leonard and Frye, 1978; Frye et al., 1978; Thomasson, 1979; Winkler, 1985; Schultz, 1990; Swisher III, 1992; Gustavson, 1996; Tedford, 1999; Tedford et al., 2004; Cepeda and Perkins, 2006; Smith et al., 2016, 2018). Due to the nature of the largely fluvial and eolian deposits, the Ogallala exhibits substantial heterogeneity north-to-south. Consequently, different workers have classified the Ogallala Formation as a Group (in the northern Great Plains) (Tedford et al., 2004) or as a formation, primarily in Kansas and to the south (Gustavson, 1996; Ludvigson et al., 2009). Herein, we refer to the Ogallala exclusively as a formation. However, in the northern Great Plains, there are further distinct formations such as the Valentine, Ash Hollow and Olcott Formations (Joeckel et al., 2014; Smith et al., 2017), each with defined age constraints based upon biostratigraphy and ashes (Tedford et al., 2004). In the southern Great Plains, previous workers have proposed elevating the Ogallala Formation to group status, based upon subdividing the Ogallala into the Bridwell and Couch Formations (Winkler, 1985; Gustavson and Winkler, 1988). However, in the southern Great Plains, we adopt the terminology of Gustavson (1996) who concluded that these formations are difficult to map and contain little dateable material, suggesting that the Ogallala remain with formation status. The thickness of Ogallala sediments generally varies relative to the underlying topography between 250 m, in regions where it fills paleovalleys, and 10-30 m in the interfluves between paleovalleys (Gustavson and Holliday, 1999). In the southern Great

Plains, there is frequently an erosion-resistant caliche or caprock calcrete that separates the Ogallala Formation from the pre-
200 dominantly eolian Plio-Pleistocene the Blackwater Draw Formation and, locally, the lacustrine Blanco Formation (Gustavson, 1996; Gustavson and Holliday, 1999). These caliche caprocks are thought to have developed during one or multiple periods of extended landscape stability and likely record a multi-genetic history (Brock and Buck, 2009; Henry, 2017). In contrast, in the northern Great Plains, the Ogallala Formation is overlain by several high energy deposits, including the Crooked Creek Formation in Kansas and the Broadwater Formation in Nebraska (Swinehart and Diffendal Jr., 1987).

To capture spatial changes in precipitation $\delta^{18}O$—and hence shifts in the aridity gradient—we collected samples spanning nearly the entire N-S and E-W extent of the Ogallala Formation from paleosol authigenic carbonate material to reconstruct paleo-precipitation $\delta^{18}O$. We build upon previous work (Fox and Koch, 2003, 2004) that developed a spatially-extensive dataset of paleosol carbonate isotopes, collected primarily to understand changes in C3/C4 vegetation during the late Miocene. We build upon these datasets, focusing on filling gaps in the southern and southwestern Great Plains (Texas and New Mexico),
while also contributing additional data in the central and northern Great Plains.

Though the Ogallala Formation provides an unparalleled opportunity to collect spatially extensive $\delta^{18}O_c$ data, the precise chronology of Ogallala deposition remains uncertain. In many places, specific formations have been dated using biostratgraphy and radiometrically dated ashes (Tedford et al., 2004); however in many other places, particularly those Ogallala outcrops to the west and disconnected from the primary exposure of the Ogallala, temporal constraints are provided by lithologic correlations
(Frye et al., 1982). Even in the formations which have been dated using biostratigraphy, the age constraints are relatively broad, typically limited by the precision of the North American Land Mammal Ages. Lastly, the relatively thin veneer of Ogallala sedimentation combined with the potentially long timespan covered by deposition suggests that there are frequent and temporally extensive unconformities within many sections (Smith and Platt, 2023). As a consequence, it remains difficult to correlate sections across the large expanse of the Great Plains.

For the purposes of our study, we use the published age constraints (Table 1) from either the study that originally studied in detail the sampled section or from later publications that provide a more precise age. Because we are only interested in broadly comparing samples spatially, we treat all samples in a given section as having the same age. We bound this age by considering the maximum possible age range for the given formation, which is either the Ogallala Formation or, in the northern Great Plains, one of the formations witin the Ogallala Group. For example, in locations where we sample well-defined and
dated subdivisions of the Ogallala Group (for example, samples from the Ash Hollow Formation at Lake McConaughy State Recreation Area in Nebraska (Joeckel et al., 2014)), we consider our sample ages to be bound by the full age range of the formation. In other cases, we know only that our data lies above or below a certain dated ash (for example, samples from Wildcat Bluff Nature Center in Texas (Cepeda and Perkins, 2006)) and therefore, one bound on the age is provided by the dated material. In some cases, the sampled unit has only been correlated to the Ogallala Formation and there are no other age
constraints to narrow the large possible range of ages. We therefore adopt the full possible range of ages given the identification of the unit as the Ogallala Formation. For example, for many sections in New Mexico, Ogallala outcrops have been identified by lithologic or geomorphic correlation to the contiguous body of the Ogallala Formation to the east, but no dateable material has been recovered (Frye et al., 1982). Despite these broad age constraints, we find that our results are not sensitive to the

uncertainty in our correlations nor to the precise chronology of Ogallala deposition because $\delta^{18}O_c$ is largely invariant within any given section.

Lastly, we also collected samples from the Miocene-age Tesuque Formation within the Santa Fe Group within the Rio Grande Rift. These samples are the only samples we collected west of the original spatial extent of the Ogallala Formation. Unlike the Ogallala Formation, the Santa Fe Group comprises thick basin fill shed off the Sangre de Cristo Range during ongoing rift extension (Galusha and Blick, 1971; Kuhle and Smith, 2001) and chronological constraints are provided by a combination of magnetostratigraphy (Barghoorn, 1981, 1985), biostratigraphy (Galusha and Blick, 1971; Tedford and Barghoorn, 1993; Aby et al., 2011), and radiometric dates (Mcintosh and Quade, 1996; Izett and Obradovich, 2001; Koning et al., 2013). The Tesuque Formation contains abundant authigenic carbonates, including nodules, root casts, groundwater cements and laterally extensive Bk horizons (Kuhle and Smith, 2001). To place our samples within an existing stratigraphic framework, we collected samples along the Arroyo de los Martinez section and refer readers to Koning et al. (2013) for a detailed description of this section.

## 4  Methods

### 4.1  Stable isotope measurements

We collected carbonates (n = 344) from paleosols from 32 distinct sites within the Ogallala Formation or from the Tesuque Formation. In every section, we sampled a wide-variety of carbonate types, including rhizoliths, nodules, burrows, carbonate-cemented matrix samples, and caliches in order to test whether these different carbonate types reveal different spatial isotope patterns. Individual sample types for each sample are listed in Table S1 in Manser et al. (2023). During sampling, we first removed the weathered surface layer before selecting samples. Sampling sites are shown in Fig. 1a, sample types are depicted in Fig. 2, and coordinates are reported in Table 1. Samples were powdered for isotopic analysis with a Dremel or crushed with a mortar and pestle to obtain a homogeneous powder. Before powdering nodules, caliches, and matrix samples, any outer weathered rind was first discarded. Carbon and oxygen isotope ratios of the carbonate were measured with a ThermoFisher Gas Bench II coupled via a ConFlow IV interface to a Delta V Plus mass spectrometer at ETH Zürich following procedures described in detail in Breitenbach and Bernasconi (2011). Briefly, 140-300 $\mu$g of the powdered sample, depending on carbonate content, were reacted with 5 drops of 104% phosphoric acid at 70 ℃ in He-flushed exetainers. Each batch of 79 samples included 16 replicates of the internal standards MS2 ($\delta^{13}C$ = +2.13 ‰, $\delta^{18}O$ = -1.81 ‰) and ETH-4 ($\delta^{13}C$ = -10.19 ‰, $\delta^{18}O$= -18.71 ‰) insterspersed throughout the run. The standards are used for drift corrections and data normalization and are calibrated to the international reference materials NBS 19 ($\delta^{13}C$ = +1.95 ‰, $\delta^{18}O$= -2.2 ‰) and NBS 18 ($\delta^{13}C$= -5.01 ‰, $\delta^{18}O$= -23.00 ‰; Bernasconi et al., 2018). Analytical reproducibility of the standards was better than 0.1 ‰ ($1\sigma$ for both $\delta^{13}C$ and $\delta^{18}O$). We convert our $\delta^{18}O_c$ data from VPDB to VSMOW using the equations in Brand et al. (2014), and we report all $\delta^{18}O_c$ data relative to VSMOW.

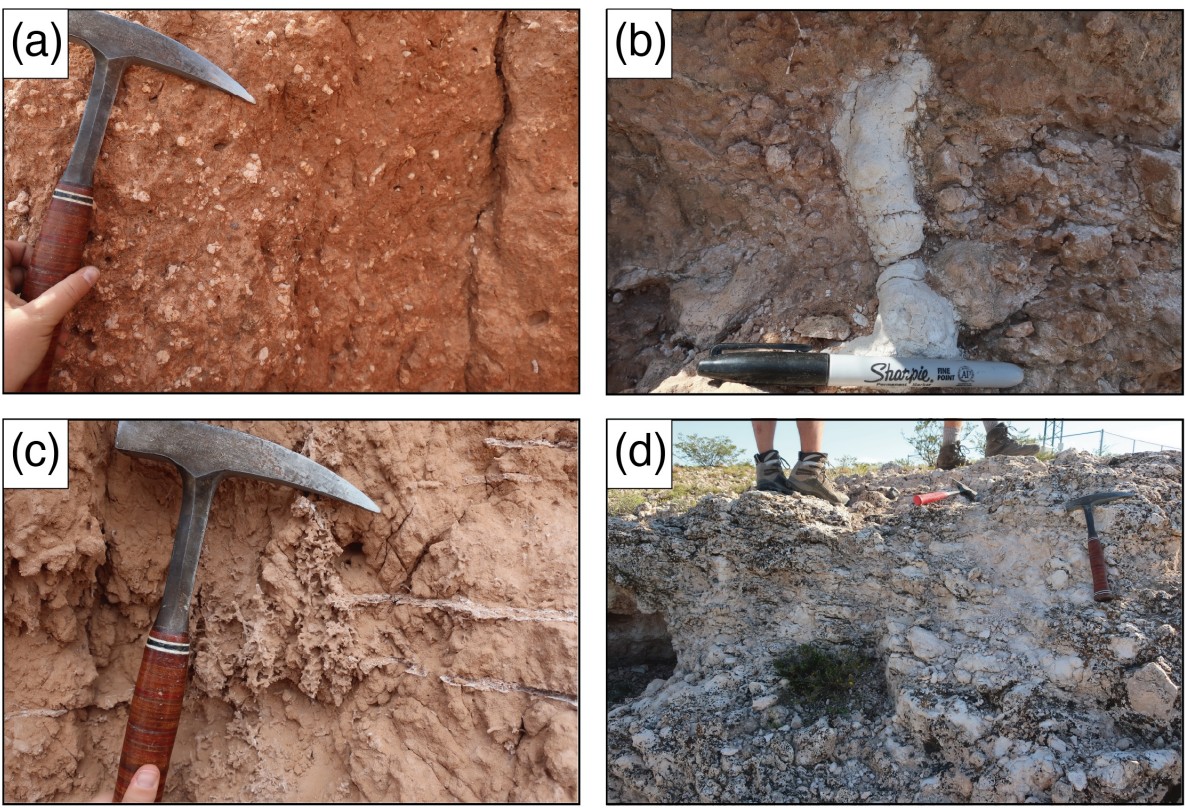

**Figure 2.** Field photos representing the primary types of authigenic carbonate sampled in this study. (a) Carbonate nodules. (b) Burrows. (c) Root casts. (d) Caprock calcrete, as pictured in SE New Mexico.

## 4.2 HYSPLIT

Because $\delta^{18}O_p$ is heavily influenced by the pathway that moisture takes to reach a certain site, we use NOAA's Hybrid Single-Particle Lagrangian Trajectory Model (HYSPLIT) (Draxler and Hess, 1998; Stein et al., 2015) to analyze the pathways by which moisture reaches the Great Plains. HYSPLIT is commonly used to understand both modern precipitation $\delta^{18}O_p$ data (Sjostrom et al., 2006; Bershaw et al., 2012; Li and Garzione, 2017; Zhu et al., 2018; San Jose et al., 2020) and to yield insights into the controls on reconstructed paleo-precipitation $\delta^{18}O$ (Oster et al., 2012; Lechler and Galewsky, 2013; Caves et al., 2014, 2015; Wheeler and Galewsky, 2017; Zhu et al., 2018). To track air parcels back in time and space from a given location, we use NARR reanalysis data which has a 32x32 km resolution (Mesinger et al., 2006) as the HYSPLIT climatological model input. To resolve spatial variability in the origins and pathways of storms, we simulate the origin and pathway of the air parcels for four selected sites (32° N & -97° E; 32° N & -105° E; 42° N & -97° E; 42° N & -105° E). At each site, we initialize the air parcel at 1000 m above ground. We choose this height as this level encapsulates much of the bulk moisture transported by the GPLLJ, which has maximum wind speeds between 500-1000 m above ground level (Jiang et al., 2007), and also captures

moisture transport by the mid-latitude westerlies (Lechler and Galewsky, 2013; Wheeler and Galewsky, 2017). The sites were chosen to encapsulate nearly the full latitudinal and longitudinal range represented by the carbonate stable isotope data. At each site, we generate nearly 53,000 back-trajectories (i.e., one trajectory every 6 hours from 1980-2016) and filter these trajectories for only those that are estimated to produce precipitation within 6 hours of reaching the endpoint (e.g. Lechler and Galewsky,

2013; Caves et al., 2015). This results in approximately 4,000 to 11,000 trajectories at each site. We further use HYSPLIT's built-in clustering algorithm to calculate the percentage of trajectories that originate from the Gulf of Mexico. Lastly, we note two critical assumptions regarding this HYSPLIT analysis. First, we assume that, by tracking air-parcels using HYSPLIT, we are also tracking moisture; however, HYSPLIT takes does not account for moisture addition by evaporation or removal by precipitation and does not track diffusion of moisture into and out of an air mass. The assumption of moisture transport by

advection is sometimes violated in regions of strong air-mass mixing, such as in the Great Plains (Draxler and Hess, 1998); however, comparison of HYSPLIT results with the results of a more rigorous moisture tracking model—the Water Accounting Model (WAM-2layers)—generally shows close agreement (Driscoll, 2022). Second, our HYSPLIT results are strictly only applicable to understand the modern climate. Nevertheless, we use these results to develop insights into the controls on past precipitation $\delta^{18}O$.

### 4.3 Vapor transport model

Given the moisture pathways predicted by HSYPLIT, we use a one-dimensional vapor transport model (Kukla et al., 2019) to predict the isotopic composition of precipitation transported along these pathways. This model links spatial patterns of $\delta^{18}O_p$ to the balance of three moisture fluxes—precipitation (P), evapotranspiration (ET), and transport. It uses energetic and mass balance limits on evapotranspiration to place constraints on the relationship between P and ET, which is a key parameter that

controls spatial patterns of $\delta^{18}O_p$ (Salati et al., 1979; Gat and Matsui, 1991; Lee et al., 2007; Winnick et al., 2014; Bailey et al., 2018; Kukla et al., 2019). Together with the dry and moist adiabatic lapse rate ($\Gamma$ [°K/m]) and an assumed environmental lapse rate ($\gamma$ [°K/m]), orographic rainout is incorporated into the model following the work of Smith (1979) and Smith and Barstad (2004). Though there are a variety of topographic parameterizations, for our simulations of westerly-derived moisture we use an idealized topography with a Gaussian-shaped mountain range combined with an orogenic plateau in the lee of the

range. In our simulations of GPLLJ $\delta^{18}O_p$, we assume flat terrain. Even though the Plains gently rise by more than 1000 m from the Gulf Coast to the Front Range, the model of orographic precipitation applies to adiabatic ascent over topography—a process that does not occur over the Plains. Equation 1 is used to calculate the column-integrated precipitable water content ($w$ [kg/m$^2$]) as a function of advection ($u$ [m/s], the movement of an air parcel) and eddy diffusion, precipitation ($P$ [kg/m$^2$]) and evapotranspiration ($ET$ [kg/m$^2$s]) (Kukla et al., 2019).

$$\frac{\partial w}{\partial t} = \nabla \cdot (D \nabla \cdot w) - u \nabla \cdot w + ET - P \tag{1}$$

where $t$ is time and $D$ is the coefficient for eddy diffusion (m$^2$/s).

Equation 2 is used to calculate the isotopic ratio of precipitable water ($r_w$) over space ($x$) in steady state following equation 1. The isotope ratio of precipitation ($r_P$) is derived by assuming equilibrium fractionation during moisture condensation. The isotope ratio of ET ($r_{ET}$) is derived as a function of the transpired fraction of ET (T/E+T) and the balance of equilibrium and kinetic isotope fractionation. For further details, we refer the reader to Kukla et al. (2019).

$$0 = D(\frac{d^2 r_\mathrm{w}}{dx^2} + 2\frac{1}{w}\frac{dw}{dx}\frac{dr_\mathrm{w}}{dx}) - u\frac{dr_\mathrm{w}}{dx} + \frac{ET}{w}(r_\mathrm{ET} - r_\mathrm{w}) - \frac{P}{w}(r_\mathrm{p} - r_\mathrm{w}) \tag{2}$$

Here, $r$ refers to the $^{18}O/^{16}O$ ratio and subscripts $ET$, $P$, and $w$ refer to evapotranspiration, precipitation, and precipitable water, respectively.

Equations 1 and 2 ensure mass conservation and permit the reduction of precipitable water and of the $^{18}O$ of precipitable water by precipitation. Mass conservation relationships between potential evapotranspiration (PET), evapotranspiration (ET) and precipitation (P) are encapsulated within a Budyko hydrologic balance framework that limits moisture recycling based on energy (when PET $\leq$ P) or water (when P $\leq$ PET). Equation 3 can be used to calculate a so-called "Budyko curve", which determines a unique hydroclimate solution for each isotope gradient using a given value of $\omega$——a non-dimensional free parameter that captures land surface characteristics (such as vegetation, bedrock lithology, ruggedness, etc.) and modulates the partitioning of precipitation into either runoff or evapotranspiration (Budyko, 1974; Fu, 1981; Zhang et al., 2004; Greve et al., 2015; Kukla et al., 2019).

$$ET = P\left(1 + \frac{PET}{P} - \left(1 + \left(\frac{PET}{P}\right)^{\omega}\right)^{\frac{1}{\omega}}\right) \tag{3}$$

To populate the parameters in this model, we again use North American Regional Reanalysis (NARR) data (long-term monthly mean for years 1979-2000) (Mesinger et al., 2006). We show input values for model parameters in the appendix (Table 2). For $\omega$, we use the global mean value of 2.6 (Greve et al., 2015). For the transpired fraction of ET (T/E+T) we use a value of 0.64 (Good et al., 2015).

We simplify the longitudinal differences in simulated GPLLJ trajectories (see Fig. 3) as a 1-D storm track that transports moisture from the Gulf of Mexico to the Great Plains. To reflect the general curvi-linear trajectories of the GPLLJ, partly caused by the influence of the North American Cordillera (Jiang et al., 2007), we implement a bend in our simplified trajectories at 32° N. From this bend, the simulated trajectory runs along the 101° meridian. This trajectory ends at 43° N, the latitude of our northernmost site. This trajectory passes over the middle of the present-day exposure of the Ogallala Formation and, hence, captures a representation of the atmospheric processes that result in $\delta^{18}O_p$ over the Great Plains.

To test for the influence of westerly moisture in the Great Plains (as indicated by the HYSPLIT results), we initialized the vapor transport model with annual mean NARR data interpolated to a simplified 1-D trajectory representing the Westerlies (Figure 8). The trajectory latitude is chosen to lie in between the northern and southern boundary of the Ogallala Formation. To account for the known effects of orographic rainout on westerly moisture (Friedman et al., 2002; Lechler and Galewsky, 2013;

Mix et al., 2019), we use a simplified topography, based on the modern observed topographic profile, that follows a Gaussian shaped mountain with a flat plateau in the lee.

## 5   Results

### 5.1   Moisture sources and precipitation trajectories

Contour plots in Figures 3a-d show the percentage of trajectories that produce precipitation at 1000 m above ground level for each of the plotted locations over the course of a year. Western sites receive a substantial portion of their moisture from westerly or southwesterly trajectories that traverse the high topography of the North American Cordillera. In particular, the northwestern-most site (Figure 3a), receives little moisture from the Gulf. In contrast, the southeastern-most site (Figure 3d) receives predominantly Gulf moisture. These patterns vary somewhat seasonally, with Gulf/southeasterly moisture more dominant in the spring and summer months and westerly moisture more dominant in the winter months. Our results from HYSPLIT's clustering algorithm show that northern locations receive less gulf/southeasterly moisture than southern locations; however, regardless of latitude, the percentage of storms sourced from the Gulf increases to the east (Figure 3e).

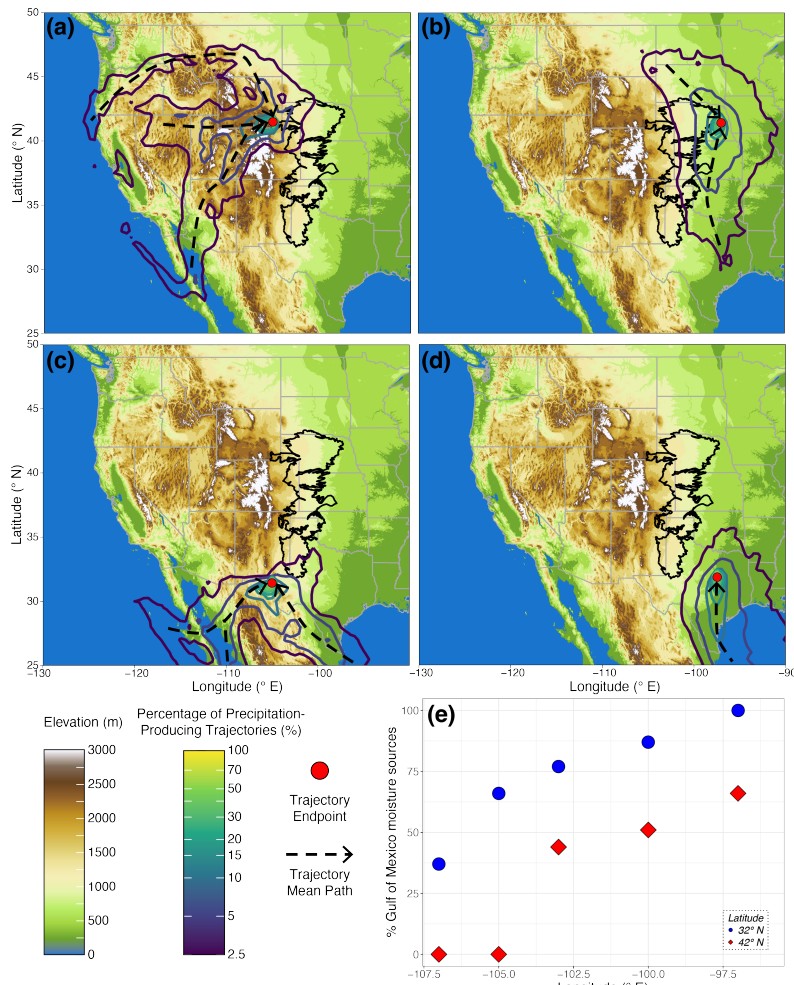

**Figure 3.** Map showing the percentage of precipitation producing storm-tracks at four localities on the Great Plains that bound the latitudinal and longitudinal extent of our data. HYSPLIT air parcels at all four sites are initialized at 1000 m above ground. (a) 42° N/-105° E. (b) 42° N/-97° E. (c) 32° N/-105° E. (d) 32° N/-97° E. The resulting mean storm-trajectories are demarcated with dashed, black lines with an arrow showing the direction of transport. In panels a-d, the extent of the Ogallala is shown in black. (e) An estimate of the percentage of Gulf/southeasterly moisture that reaches each location as a function of longitude. Red points represent a transect of HYSPLIT simulations along the 42° N parallel; blue points represent a transect of HYSPLIT simulations along the 32° N parallel.

## 5.2   $\delta^{18}O_c$ and $\delta^{13}C$ Results

Our new $\delta^{18}O_c$ data range from a minimum of 17.0 ‰ to a maximum of 27.3 ‰ (Table S1 in Manser et al. (2023)). When averaged by section, our new $\delta^{18}O_c$ data range from 19.6 ‰ to 26.8 ‰ (Table 1). The compiled data have a much larger range, reflecting their larger latitudinal distribution, with the average section $\delta^{18}O_c$ ranging from 13.6 ‰ to 27.0 ‰ (Appendix Table

**Table 1.** Site-averaged data. Values of $\delta^{13}C$ reported in ‰ relative to VPDB. Values of $\delta^{18}O$ reported in ‰ relative to VSMOW. n is number of samples collected from the site. Mean annual temperature data retrieved from NARR (Mesinger et al., 2006)

| Lat (°N) | Lon (°E) | ID | $\delta^{13}C$ | $\delta^{13}C$ 1σ | $\delta^{18}O_c$ | $\delta^{18}O_p$ | $\delta^{18}O$ 1σ | $\delta^{18}O$ range | Bottom age (Ma) | Top age (Ma) | MAT (°C) | n | Reference |
|---|---|---|---|---|---|---|---|---|---|---|---|---|---|
| 34.91 | -103.45 | BV | -4.64 | 1.86 | 25.36 | -5.15 | 0.45 | 1.80 | 23 | 5 | 15.35 | 18 | Gustavson 1996 |
| 40.98 | -103.47 | CB1 | -6.69 | 0.36 | 21.86 | -9.70 | 0.67 | 2.78 | 31.8 | 5 | 10.91 | 22 | Galbreath 1953; Tedford 1999, 2004 |
| 40.92 | -103.29 | CB2 | -6.27 | 0.18 | 22.72 | -8.82 | 0.59 | 1.23 | 31.8 | 5 | 10.99 | 5 | Galbreath 1953; Tedford 1999, 2004 |
| 34.45 | -101.11 | CC | -7.34 | 0.75 | 26.29 | -3.85 | 0.31 | 1.04 | 11.6 | 7.6 | 16.92 | 17 | Lehmann and Schnable 1992 |
| 36.57 | -103.30 | CLSP | -6.63 | 0.64 | 24.98 | -6.07 | 0.11 | 0.23 | 23 | 5 | 13.08 | 3 | Frye et al. 1978 |
| 33.41 | -103.75 | CP | -6.18 | 1.55 | 25.26 | -4.91 | 1.24 | 3.53 | 13.6 | 10.3 | 16.83 | 13 | Henry 2017 |
| 38.64 | -100.91 | DB | -6.01 | 0.56 | 23.91 | -7.08 | 0.47 | 1.90 | 11.4 | 7.5 | 13.28 | 19 | Smith et al. 2011; 2016 |
| 33.75 | -104.58 | EPB | -5.00 | 0.71 | 25.13 | -4.91 | 0.43 | 1.41 | 23 | 5 | 17.40 | 8 | Frye et al. 1982 |
| 36.02 | -105.97 | ESP | -5.93 | 1.05 | 19.81 | -12.10 | 2.12 | 7.82 | 16.2 | 14 | 9.50 | 58 | Koning et al. 2013 |
| 34.59 | -104.04 | GVR | -5.68 | 0.47 | 25.03 | -5.40 | 0.33 | 0.97 | 8.3 | 4.7 | 15.73 | 8 | Gustavson 1996 |
| 41.21 | -101.67 | MCA | -6.58 | 0.26 | 19.56 | -11.98 | 0.70 | 2.13 | 13 | 5 | 10.99 | 7 | Joeckel et al. 2014 |
| 41.20 | -101.67 | MCC | -7.00 | 0.46 | 19.61 | -11.94 | 0.85 | 2.74 | 13 | 5 | 10.99 | 16 | Joeckel et al. 2014 |
| 41.21 | -101.67 | MCE | -6.75 | 0.48 | 19.83 | -11.72 | 1.04 | 3.11 | 13 | 5 | 10.98 | 6 | Joeckel et al. 2014 |
| 36.10 | -104.26 | MI | -5.55 | 0.57 | 25.61 | -5.74 | 0.13 | 0.23 | 23 | 5 | 11.78 | 3 | Frye et al. 1978 |
| 36.10 | -104.26 | MI2 | -2.27 | 1.30 | 26.12 | -5.23 | 0.55 | 1.16 | 23 | 5 | 11.80 | 4 | Frye et al. 1978 |
| 34.98 | -101.69 | PD | -6.34 | 0.59 | 25.41 | -5.02 | 0.28 | 0.89 | 23 | 5 | 15.72 | 9 | Lucas et al. 2001 |
| 33.01 | -103.87 | PO | -6.50 | 0.80 | 25.71 | -4.36 | 0.44 | 2.01 | 11 | 4.5 | 17.29 | 20 | Gustavson 1996; Henry 2017 |
| 37.10 | -101.94 | PoR | -6.13 | 0.75 | 24.51 | -6.18 | 0.44 | 1.02 | 23 | 5 | 14.58 | 7 | Smith et al. 2015 |
| 35.99 | -103.46 | REA | -4.01 | 1.51 | 25.21 | -5.58 | 0.81 | 2.52 | 23 | 5 | 14.15 | 7 | Frye et al. 1978 |
| 35.99 | -103.46 | REA2 | -4.82 | 0.65 | 25.69 | -5.10 | 0.79 | 1.60 | 23 | 5 | 14.16 | 4 | Frye et al. 1978 |
| 35.99 | -103.46 | REA3 | -4.71 | 3.39 | 25.51 | -5.27 | 0.14 | 0.31 | 23 | 5 | 14.16 | 4 | Frye et al. 1978 |
| 34.82 | -103.75 | RG | -5.80 | 0.55 | 25.78 | -4.71 | 0.37 | 1.12 | 8.3 | 4.7 | 15.44 | 11 | Gustavson 1996 |
| 34.19 | -104.79 | RSE | -5.41 | 1.20 | 25.14 | -5.24 | 0.80 | 1.94 | 23 | 5 | 15.92 | 9 | Frye et al. 1982 |
| 39.04 | -99.54 | S9A | -7.48 | 0.67 | 25.16 | -5.88 | 0.50 | 1.62 | 11.6 | 3.6 | 13.07 | 9 | Thomasson 1979 |
| 39.04 | -99.54 | S9A2 | -7.04 | 0.15 | 25.07 | -5.98 | 0.42 | 1.01 | 11.6 | 3.6 | 13.07 | 5 | Thomasson 1979 |
| 36.67 | -103.07 | SNE | -5.47 | 0.17 | 24.71 | -6.11 | 0.06 | 0.12 | 23 | 5 | 14.01 | 5 | Leonard and Frye 1978 |
| 36.67 | -103.07 | SNE2 | -4.41 | 1.34 | 25.00 | -5.82 | 0.28 | 0.48 | 23 | 5 | 14.01 | 3 | Leonard and Frye 1978 |
| 33.43 | -101.41 | SQ3 | -6.90 | 0.49 | 26.17 | -3.87 | 0.51 | 1.46 | 10.3 | 4.9 | 17.40 | 7 | Henry 2017; Fox and Koch 2004; Gustavson 1996 |
| 33.47 | -101.51 | SQ4 | -6.15 | 1.88 | 26.76 | -3.30 | 0.32 | 0.80 | 13.6 | 10.3 | 17.31 | 4 | Henry 2017; Fox and Koch 2004; Gustavson 1996 |
| 35.24 | -101.95 | WC | -8.57 | 1.96 | 23.10 | -7.37 | 4.39 | 7.69 | 10 | 5 | 15.52 | 3 | Cepeda and Perkins 2006 |
| 34.47 | -101.11 | WK | -6.78 | 0.71 | 26.37 | -3.77 | 0.25 | 0.99 | 11 | 4.5 | 16.92 | 19 | Gustavson 1996; Gustavson and Holliday 1999 |
| 34.60 | -106.08 | WW | -3.16 | 1.00 | 24.88 | -6.43 | 0.21 | 0.59 | 23 | 5 | 11.96 | 9 | Frye et al. 1982 |

3). Within individual sections, there is very little variance: The mean range and standard deviation of $\delta^{18}O$ in individual sections is ±1.8 ‰ and ±0.68 ‰, respectively, indicating that temporal shifts in $\delta^{18}O_c$ in the sampled sections are small (Table 1; Fig. 4b). If we exclude data outside Ogallala Formation, the mean range and standard deviation of $\delta^{18}O_c$ in individual sections is lower, at ±1.7 ‰ and ±0.6 ‰, respectively, indicating that carbonates in the Ogallala Formation have very little variance.

Our new Ogallala $\delta^{13}C$ data has a mean value of -6.14 ±1.4 (1σ) ‰ (Table 1) (excluding data from the Tesuque Formation). In contrast, our compiled Ogallala Formation data has a mean $\delta^{13}C$ value of -6.74 ±0.8 (1σ) ‰. This discrepancy is due to relatively high $\delta^{13}C$ values in the sections in New Mexico identified via lithologic or geomorphic correlation by Frye et al. (1982) to be Ogallala outcrops. When these Frye et al. (1982) data are excluded, the mean $\delta^{13}C$ value in our new data is -6.69 ±0.9 (1σ) ‰, which is statistically indistinguishable from the compiled data $\delta^{13}C$ mean.

## 5.3 Spatial distribution of $\delta^{18}O_p$

For each site, we calculate a mean reconstructed $\delta^{18}O_p$ value and compare these mean values with modern water $\delta^{18}O$ (retrieved from the waterisotopes database (Waterisotopes Database, 2019)). To reconstruct paleo-precipitation $\delta^{18}O$, we use
the modern $2\,\mathrm{m}$ air-temperature from NARR reanalysis to estimate the formation temperature of soil carbonate along with
the fractionation factors of Kim and O'Neil (1997). Means, standard deviations, and full $\delta^{18}O$ ranges for each site with new
data is presented in Table 1 and the previously published $\delta^{18}O$ data—converted to paleo-precipitation $\delta^{18}O_p$—is presented in
Appendix Table 3. All of the $\delta^{18}O_c$ data is available as a supplementary file (Table S1) in Manser et al. (2023). The spatial
distribution of the reconstructed $\delta^{18}O_p$ shows a nearly identical spatial distribution to that of modern $\delta^{18}O$ (Fig. 4a). From the
Gulf Coast inland, paleo $\delta^{18}O$ follows the modern $\delta^{18}O$ gradient, with lower values to the west and to the north. The maximum
change in both paleo and modern $\delta^{18}O$ occurs along a southeast-northwest trend, running roughly from the Gulf Coast in Texas
to northern Colorado.

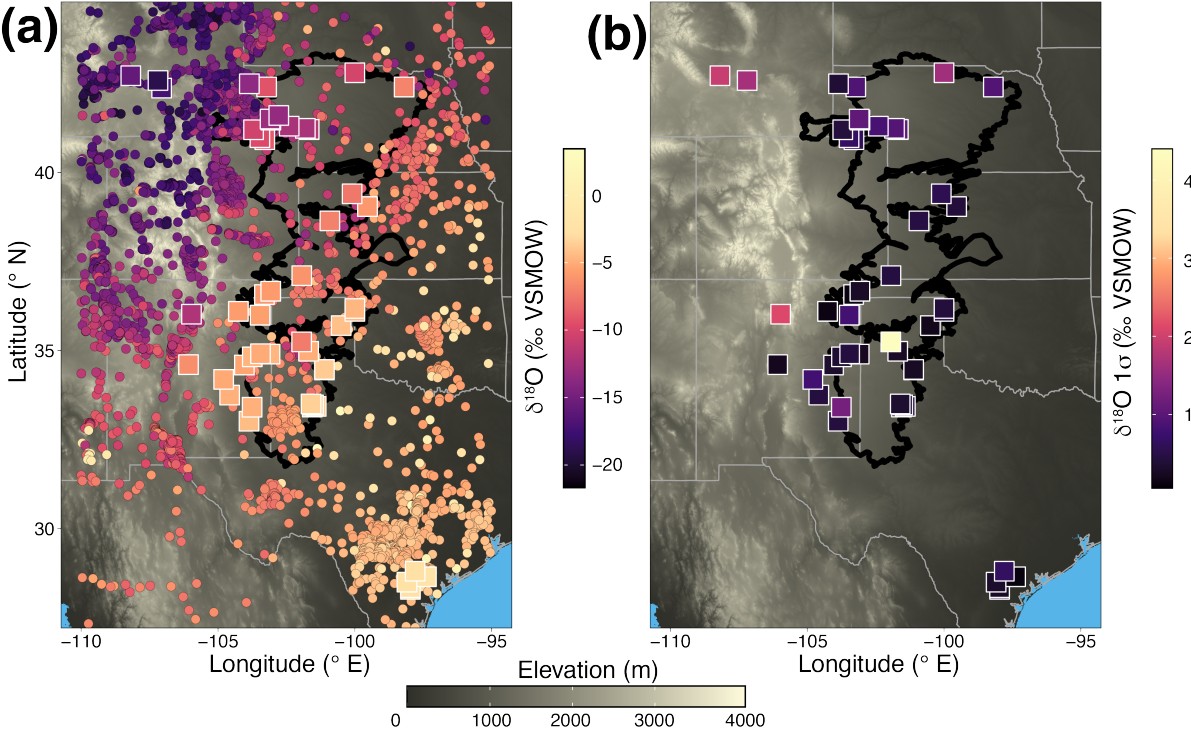

**Figure 4.** Spatial distribution of $\delta^{18}O$ (a) and reconstructed $\delta^{18}O_p$ $1\sigma$ (b). (a) Comparison of reconstructed $\delta^{18}O_p$ from $\delta^{18}O_c$ (large squares) calculated using the yearly mean of monthly long-term temperature data retrieved from NARR for each site. Modern meteoric water $\delta^{18}O$ (small circles) are derived from modern groundwater, river or stream water retrieved from the waterisotopes database (Waterisotopes Database, 2019). Data points are colored by their $\delta^{18}O$ values. (b) The $1\sigma$ of the reconstructed $\delta^{18}O_p$, derived from the variability in the $\delta^{18}O_c$ values from each stratigraphic section. In both panels, the black polygon marks the extent of modern-day exposure of the Ogallala Formation.

We further test whether decreasing temperatures northward significantly influences our reconstructed $\delta^{18}O_p$ values. The modern mean annual temperature gradient in the Great Plains (from 28.3° N to 43° N), is -1.2° C per 100 km. In June-July-August (JJA), this gradient is reduced to 0.4° C per 100 km. We use these two temperature gradients to calculate the effect of changing the spatial pattern of temperature on our reconstructed $\delta^{18}O_p$, as well as testing a hypothetical case with latitudinally constant temperature of 25° C. Using the JJA temperature gradient, the reconstructed $\delta^{18}O_p$ gradient is slightly shallower (red circles in Fig. 5) compared to the modern $\delta^{18}O$ gradient (grey dots) and compared to the reconstructed $\delta^{18}O_p$ gradient (blue circles) using modern yearly mean NARR temperature data (Fig. 5b). The likely maximum difference that a reduced temperature gradient—relative to the modern—can impart on our results is captured in the scenario of spatially uniform temperatures, which represents an extreme end-member scenario where poleward warming in a higher $CO_2$ world results in a negligible latitudinal temperature gradient. Even spatially uniform temperatures only result in an increase of 2.9‰ in the northernmost sites relative to the assumption of a modern temperature gradient, still within the range of modern $\delta^{18}O$ across the Plains. These results suggest that our reconstructed $\delta^{18}O_p$ values are relatively insensitive to assumptions of carbonate formation temperatures in a warmer world with a potentially reduced meridional temperature gradient (Feng et al., 2016; van Dijk et al., 2020).

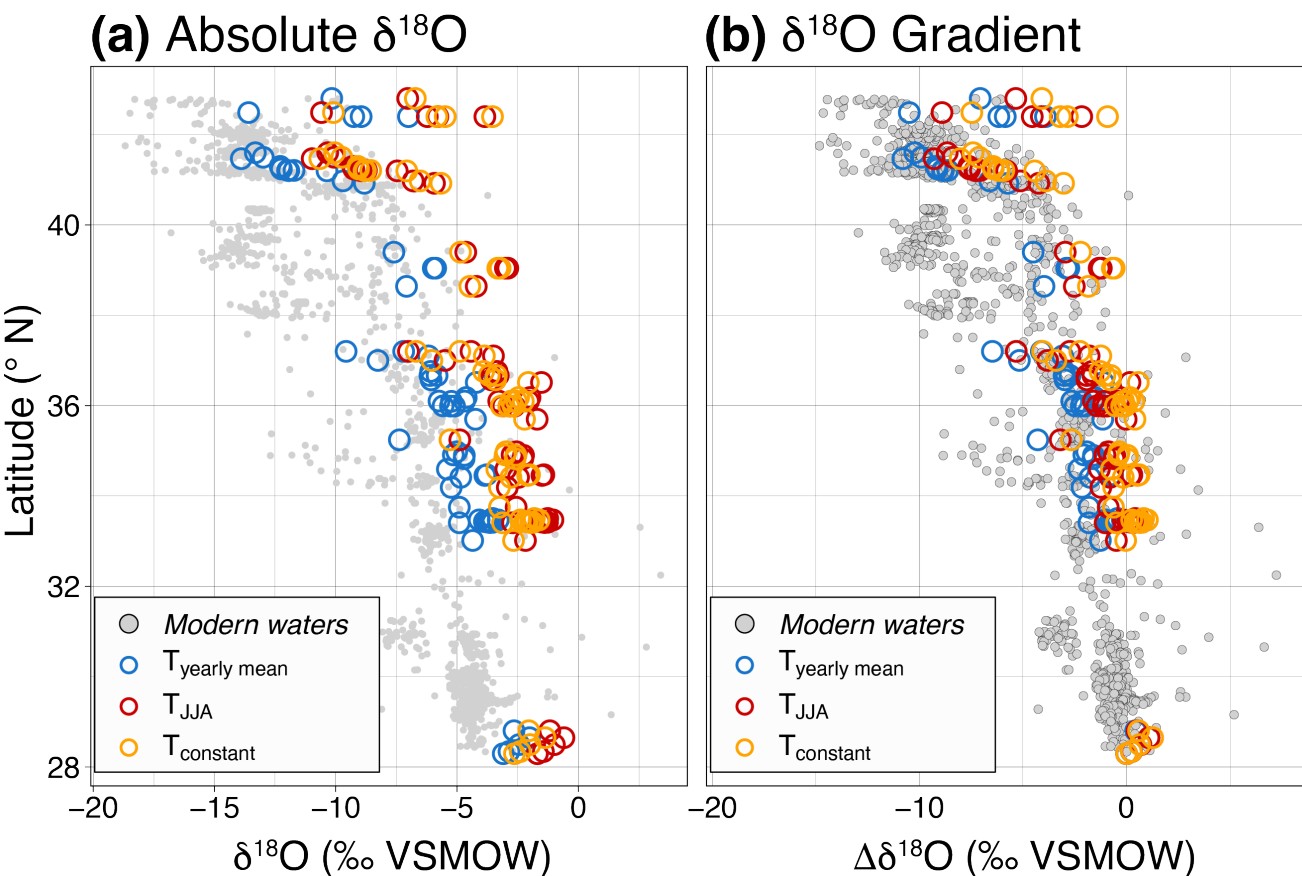

**Figure 5.** (a) Reconstructed $\delta^{18}O_p$ plotted against latitude. Reconstructed $\delta^{18}O_p$ is calculated assuming a spatially uniform temperature of 25°C (yellow circles), annual mean temperatures (blue circles), or June-July-August (JJA) mean temperatures (red circles). Annual mean and JJA temperatures are taken from monthly long-term mean data retrieved from NARR (Mesinger et al., 2006). Gray circles are modern meteoroic water $\delta^{18}O$, including groundwater and river or stream water, retrieved from the waterisotopes database (Waterisotopes Database, 2019). (b) The same as (a) but the x-axis sets the southern-most data ($\sim 28°$N) to zero to emphasize the effect on the spatial gradient in $\delta^{18}O$.

There is not only a pronounced N-S $\delta^{18}O$ gradient in the paleo data, but also substantial W-E variation. Figure 6 shows modern and reconstructed $\delta^{18}O_p$ plotted against longitude, using the NARR mean annual temperature to constrain the formation temperature of carbonate. Again, as with the comparison against latitude (Fig. 5), all reconstructed $\delta^{18}O_p$ estimates fall within the same range as the modern $\delta^{18}O$ when projected to longitude. Though again this conclusion is tempered by our assumption of carbonate formation temperatures, we suggest, as above, that this is likely to be a negligible effect, particularly given the relative insensitivity in the $^{18}O$ fractionation factor to temperature ($\sim 0.2‰$ K$^{-1}$) (Kim and O'Neil, 1997).

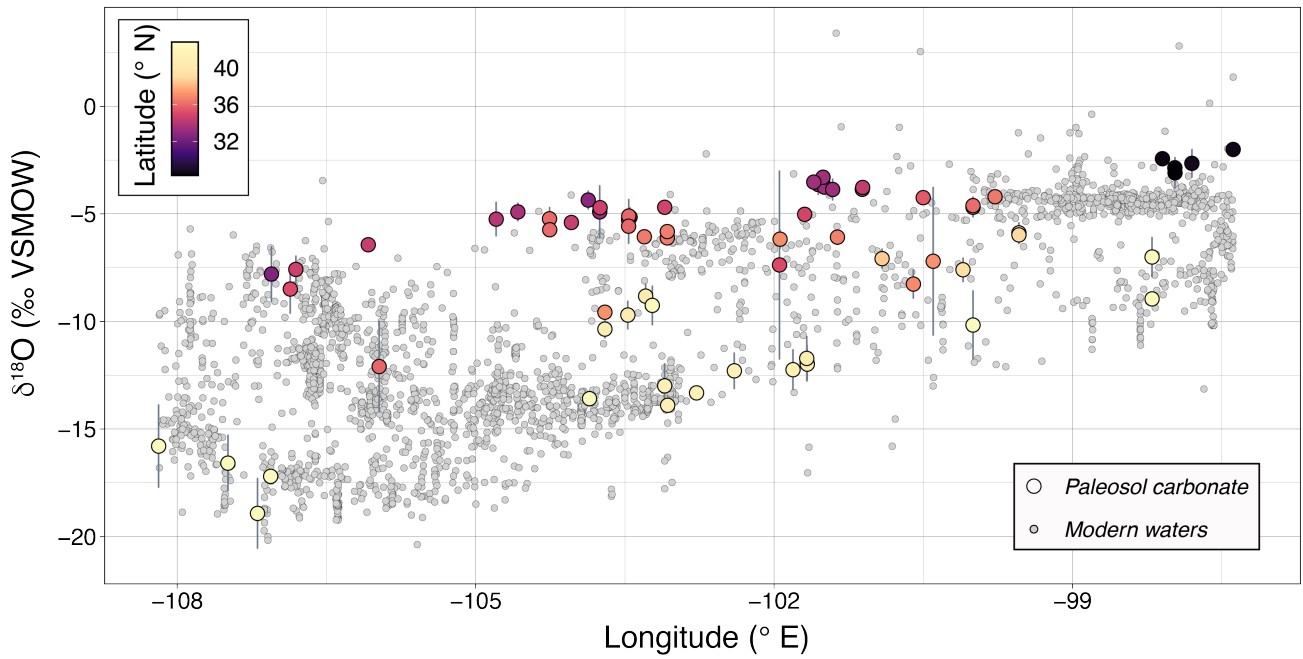

**Figure 6.** Modern meteoric water $\delta^{18}O$ (small, gray circles) and reconstructed $\delta^{18}O_p$ (large, colored circles) versus longitude. Reconstructed $\delta^{18}O_p$ is colored with respect to latitude. Modern meteoric water data includes $\delta^{18}O$ data from groundwater, river, and stream water, retrieved from the waterisotopes database (Waterisotopes Database, 2019). Vertical error bars are $1\sigma$ of the mean section $\delta^{18}O$ data.

### 5.4 Reactive transport modeling of $\delta^{18}O_p$

The vapor transport model (Kukla et al., 2019) allows us to predict $\delta^{18}O_p$ along a given storm track, reflecting the balance
between transport, precipitation, and evapotranspiration. Our model of GPLLJ $\delta^{18}O_p$ simulates higher $\delta^{18}O$ at the end of the trajectory compared to both modern and reconstructed $\delta^{18}O_p$ values (Fig. 7a). Consistent with higher $\delta^{18}O$ values, the model also predicts higher atmospheric moisture content and relative humidity than indicated by NARR, suggesting it is under-predicting rainout from air masses that reach the northern plains or that the single air-mass assumption implicit in this model is not valid in this region (Fig. 7b).
Our model of westerly moisture $\delta^{18}O_p$ suffers from a similar mis-match between simulated $\delta^{18}O_p$ and both modern $\delta^{18}O$ and reconstructed $\delta^{18}O_p$ (Fig. 8a). Though our westerly trajectory uses an idealized topography to simulate the topography of the North American Cordillera, we are able to approximately reproduce the decrease in $\delta^{18}O_p$ that occurs close to the coast of North America. Variations in simulated $\delta^{18}O_p$ after the initial orographic rainout are primarily driven by temperature that varies with elevation in the NARR data. This leads to minor discrepancies where elevation decreases $\delta^{18}O_p$ in modern waters, but
increases $\delta^{18}O_p$ in the model due to colder temperatures raising $\delta^{18}O_p$. Nevertheless, simulated $\delta^{18}O_p$ approximates meteoric water $\delta^{18}O$ along the westerly trajectory until this trajectory reaches the Great Plains at a distance of around 1550 km. At

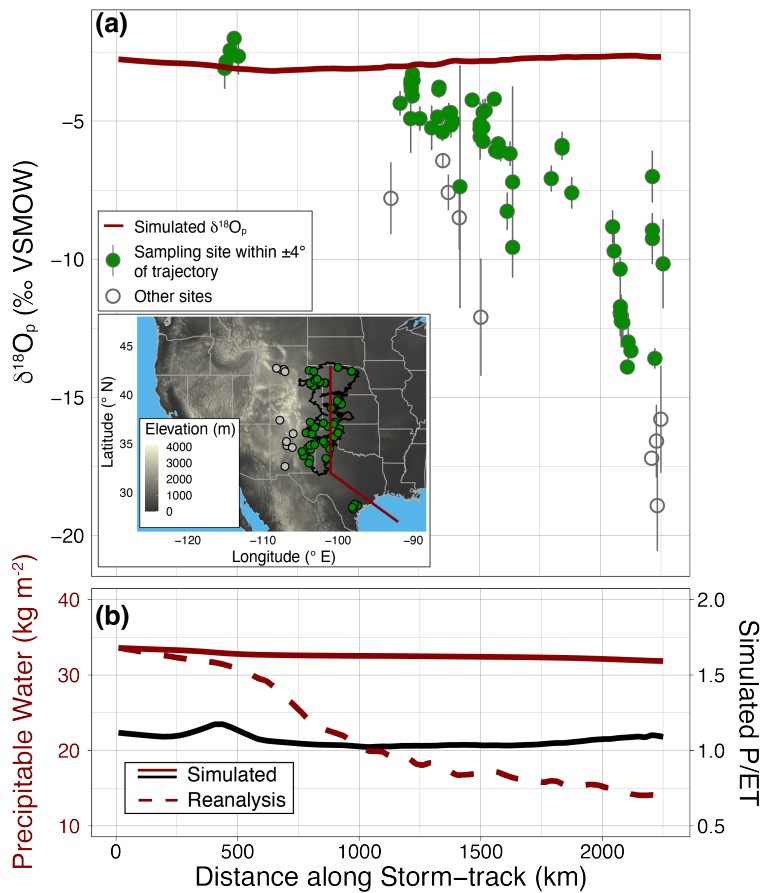

**Figure 7.** Vapor transport model simulation of $\delta^{18}O_p$ along an idealized Great Plains Low-Level Jet (GPLLJ) trajectory. (a) Red line is the modeled $\delta^{18}O_p$ along the simulated storm track (exact location of storm track shown as a red line in the inset). Green points are reconstructed $\delta^{18}O_p$ within ±4°longitude of the trajectory. Gray circles are data greater than ±4°longitude from the simulated trajectory. Vertical error bars are $1\sigma$ of the mean section reconstructed $\delta^{18}O_p$. (b) Simulated precipitable water (solid red) and P/ET (solid black) plotted along the simulated storm track. Dashed line shows the actual annual mean precipitable water along this storm track retrieved from NARR.

distances larger than this, the model substantially underestimates meteoric water $\delta^{18}O$ and reconstructed $\delta^{18}O_p$ on the Great Plains. Again examining climatological measures related to $\delta^{18}O_p$, there is a distinct discrepancy in atmospheric moisture content between the modeled output and the NARR reanalysis data over the Great Plains at a distance of around 1550 km from the starting point of the simulated westerlies trajectory (Figure 8b).

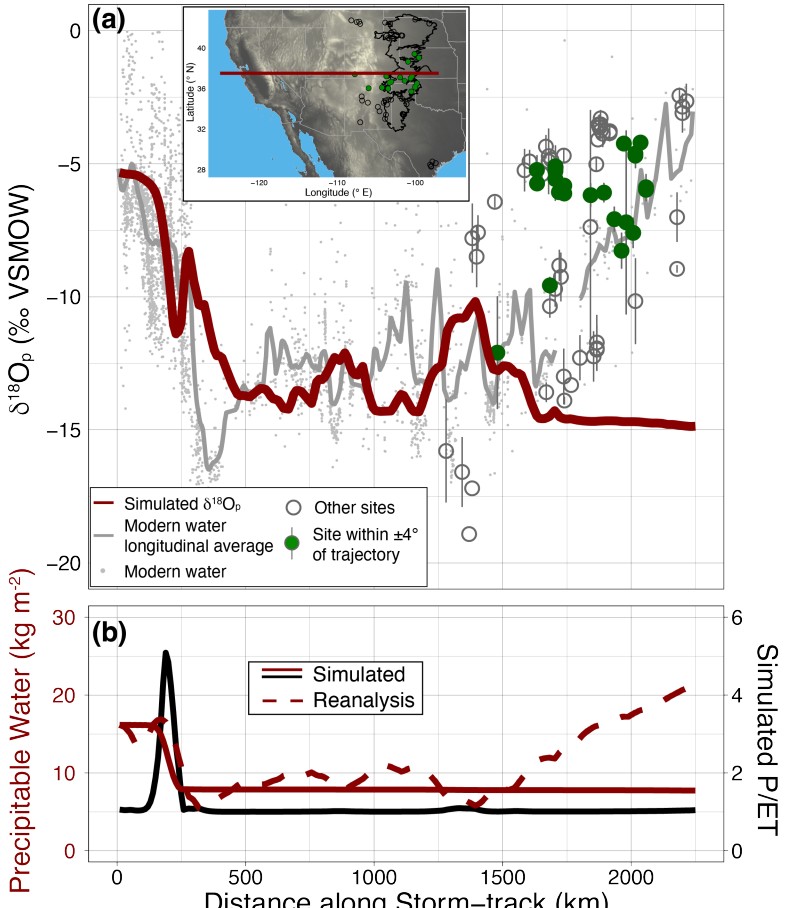

**Figure 8.** Vapor transport model simulation of $\delta^{18}O_p$ along an idealized westerly trajectory. (a) Red line is the modeled $\delta^{18}O_p$ along the simulated storm track (exact location of storm track shown as a red line in the inset). Green points are reconstructed $\delta^{18}O_p$ data within $\pm4°$latitude from the storm track. Large, unfilled gray circles are data greater than $\pm4°$latitude from the simulated trajectory. Vertical error bars are $1\sigma$ of the mean section reconstructed $\delta^{18}O_p$. Small gray circles are modern meteoric water $\delta^{18}O$, including stream, river, and groundwater, retrieved from the waterisotopes.org database (Waterisotopes Database, 2019). Gray line is a kernel-smoothed average of the meteoric water $\delta^{18}O$ against distance along the storm-track. (b) Simulated precipitable water (solid red) and P/ET (solid black) plotted along the simulated storm track. Dashed line shows the actual annual mean precipitable water along this storm track retrieved from NARR.

## 6 Discussion

Our new spatially-resolved data provide insight into moisture transport to the Great Plains during the late Neogene. In brief, the overall constancy of $\delta^{18}O$ between the late Miocene and the present suggests that the features of atmospheric circulation most responsible for moisture delivery to the Great Plains today—notably the Great Plains Low-Level Jet and the winter-time westerlies—have likely been the dominant features since at least the late Miocene. Further, that the reconstructed $\delta^{18}O_p$ gradient is indistinguishable from today's $\delta^{18}O$ gradient indicates that the balance of GPLLJ and westerly moisture has hardly changed since the late Miocene. Either these circulation features have undergone no substantial change in net rainout or, if one has, its effect was masked by countervailing changes in the other. Overall, these results bolster earlier findings by Fox and Koch (2004), who found a consistent south-to-north gradient in $\delta^{18}O_c$ in the Great Plains that they attributed to a similar latitudinal temperature gradient in the Miocene.

Below, we discuss our findings in more detail, place these data in the context of previous work in the region, and discuss how our new data permits a re-interpretation of the controls on long-term climate in the Great Plains. We also discuss several important caveats in our data that point towards the need for future work to study more nuanced changes in climate than can be resolved with this dataset.

### 6.1 Constancy of $\delta^{18}O_p$ in relation to climate change

The Miocene was warmer than present-day, particularly during the Miocene Climate Optimum (MCO) (Westerhold et al., 2020; Steinthorsdottir et al., 2021) and was characterized by a long-term cooling trend after the MCO (Herbert et al., 2016). Even after this cooling interval, the lack of extensive Northern Hemisphere ice sheets and likely a smaller Antarctic ice sheet indicate that the late Miocene was warmer than today, with a reduced latitudinal temperature gradient (LaRiviere et al., 2012; Feng et al., 2016). The warmer climate and shallower temperature gradient could impact $\delta^{18}O_p$, yet the decrease in reconstructed $\delta^{18}O_p$ with latitude appears identical to today (Fig. 4). This result is perhaps not surprising, since temperature appears to have only a secondary effect on the latitudinal gradient of $\delta^{18}O_p$ (Fig. 5a). Instead, such constancy in $\delta^{18}O_p$ through time indicates that the mixing of westerly and southerly moisture—the primary control on the latitudinal $\delta^{18}O$ gradient—was similar to today.

The results of our vapor transport modeling supports the contention that mixing between dry, low-$\delta^{18}O$ westerly air masses and moist, high-$\delta^{18}O$ southerly air masses is required to explain the long-standing presence of this steep latitudinal gradient in $\delta^{18}O_p$. The vapor transport model adequately predicts $\delta^{18}O_p$ in westerly- or GPLLJ-dominated regions, but it performs poorly where these air-masses meet (Figs. 7 & 8). Air-mass mixing is neglected in the 1-D vapor transport model, so the model's poor performance in these regions points to mixing as an important control on the latitudinal $\delta^{18}O_p$ gradient. Notably, this region of air-mass mixing is closely tied to the east-west aridity gradient because it tracks the trade-off between drier westerly air and the wetter GPLLJ. Thus, the surprisingly static spatial $\delta^{18}O_p$ pattern across the Great Plains since the Miocene could be interpreted to reflect no change in the relative mixing of dry and moist air-masses over time, maintaining the spatial pattern of the modern aridity gradient. The presence of the North American Cordillera—likely high since the Eocene (Chamberlain et al., 2012; Mix et al., 2013)—suggests that orographic rainout on the western margin of North America is also a long-standing

feature, resulting in low-$\delta^{18}O$ moisture that is advected to the Great Plains from the west. What is more surprising is that the strength of the GPLLJ appears to be similar to today even in the warmer Miocene, given that this jet and its moisture transport are likely to be sensitive to global climate change (Cook et al., 2008; Zhou et al., 2021a).

Overall, our vapor transport modeling indicates that a purely southerly source of moisture would result in $\delta^{18}O_p$ values that are too high in the northern plains (Fig. 7), whereas a purely westerly source brings $\delta^{18}O_p$ values that are too low (Figure 8). To be clear, a static $\delta^{18}O_p$ gradient does not require static precipitation and evaporation rates over time. In contrast, precipitable water, P, and E likely all increased in the warmer Miocene (Held and Soden, 2006; Siler et al., 2018), but in such a way that net rainout (and thus the $\delta^{18}O_p$ fingerprint of hydrological change) did not vary substantially.

## 6.2 Additional factors that influence $\delta^{18}O_c$

There are a number of additional factors, both climatic and non-climatic, that may affect the $\delta^{18}O_c$ data, potentially decoupling $\delta^{18}O_c$ values from $\delta^{18}O_p$ and conflicting with our interpretation above. Such factors include changes in soil temperatures and evaporation and also imprecision in the chronologies of the sections we sampled. Below, we discuss these factors and why our interpretations are robust to assumptions regarding these factors.

### 6.2.1 Effect of temperature and evaporation on $\delta^{18}O_c$

Given that temperature affects the fractionation of $^{18}O$ between water and calcite (Kim and O'Neil, 1997), spatial changes in temperature may alter our reconstructed $\delta^{18}O_p$ gradient. However, we found that even extreme scenarios for changes in the spatial pattern of temperature over the Great Plains (ranging from the modern annual average latitudinal temperature gradient to a null latitudinal temperature gradient) had only a small effect on the overall latitudinal gradient of reconstructed $\delta^{18}O_p$ (Fig. 5). We therefore conclude that, while temperature change since the Miocene has likely impacted the absolute value of $\delta^{18}O_c$, it does not substantially alter our conclusion that the reconstructed $\delta^{18}O_p$ spatial pattern is similar to today.

While the spatial pattern of temperature change in the Plains does not appear to have had a major influence on our reconstructed $\delta^{18}O_p$ gradient, for any reasonable temperature scenario (annual, JJA, or spatially uniform 25° C), reconstructed $\delta^{18}O_p$ absolute values are typically higher than modern $\delta^{18}O_p$ by 2-3‰. Fox and Koch (2004) also found that $\delta^{18}O_c$ was typically higher than predicted, particularly in the southern Great Plains, and attributed this observation to both higher temperatures and $\delta^{18}O_p$. We suggest that, while the gradient of $\delta^{18}O_p$ does not appear to have substantially changed since the Miocene, the starting value of marine vapor $\delta^{18}O$ may have been slightly higher in the warmer Miocene due to elevated precipitable water, elevating $\delta^{18}O_p$ across the Great Plains (*i.e.*, van Dijk et al. (2020)).

Recent work has indicated that evaporative conditions (*i.e.,* low AI values) may elevate $\delta^{18}O_c$, thereby leading to an overestimate of $\delta^{18}O_p$ (Kelson et al., 2023). We suggest that our overall conclusion of a similar spatial pattern of $\delta^{18}O_p$ in the late Miocene is likely not overly influenced by changes in the spatial pattern of evaporation. Today, the latitudinal gradient in AI is negligible (Seager et al., 2018b); instead, the strongest gradient in AI is west-to-east (Fig. 1). Consequently, the stark latitudinal gradient in reconstructed $\delta^{18}O_p$ is unlikely to be driven by differential evaporative conditions. While higher marine vapor $\delta^{18}O$ may explain the elevated reconstructed absolute $\delta^{18}O_p$ values (2-3‰ above modern meteoric water $\delta^{18}O$), another

explanation may be pervasive evaporative enrichment of pedogenic carbonates across the Great Plains, though this evaporative enrichment does not appear to vary latitudinally.

### 6.2.2 Changes in precipitation seasonality

Soil carbonates are thought to form mostly seasonally, particularly in the warm season and perhaps as soils dry and plants senesce, causing soil $CO_2$ to decline and carbonates to form (Breecker et al., 2009; Huth et al., 2019). However, the timing of soil carbonate formation may shift as the timing of precipitation and plant productivity varies due to climate change, and such shifts have been invoked to explain a wide-range of soil carbonate records in the western US and elsewhere (Caves, 2017; Kukla et al., 2022b; Rugenstein et al., 2022). GCMs indicate that precipitation seasonality over the Great Plains will change

substantially in a warmer world, with precipitation shifting towards the spring and away from the late summer (Cook et al., 2008; Bukovsky et al., 2017; Zhou et al., 2021a). Though it is difficult to assess the timing of even modern soil carbonate formation in the Great Plains, our reconstructed Miocene $\delta^{18}O_p$ data agree most closely with modern water $\delta^{18}O$ when using mean annual temperatures. Though we have no *a priori* reason to suspect that Miocene carbonates may record mean annual temperatures, this close agreement suggests that Miocene carbonates may have formed in the shoulder seasons (*i.e.*, spring

and fall), thereby recording intermediate temperatures and, likely, a mixture of summertime GPLLJ moisture and wintertime westerly moisture.

That Miocene soil carbonates formed in the shoulder seasons would help to reconcile our findings of a largely invariant spatial pattern of $\delta^{18}O_p$ since the Miocene with previously published records that show a large (2-4‰) increase in $\delta^{18}O_c$ during the Pliocene-Quaternary in the Great Plains (Fox et al., 2012; Mix et al., 2013; Chamberlain et al., 2014). We suggest

that this discrepancy (i.e., a static late Miocene $\delta^{18}O_p$ spatial pattern, but increasing $\delta^{18}O_c$ in the Pliocene-Quaternary) likely arises due to changes in the seasonality of soil carbonate formation between the late Miocene and Quaternary. For example, as the world cooled into the Quaternary, the southward shift of the westerly jet would suppress the springtime GPLLJ, while enhancing GPLLJ precipitation over the Great Plains during the summer—the opposite response than observed in GCMs as $CO_2$ rises (Cook et al., 2008; Bukovsky et al., 2017; Zhou et al., 2021a). Such a seasonality shift might shift the timing of

soil carbonate formation towards the summer, elevating $\delta^{18}O_c$ as it records higher summertime $\delta^{18}O_p$ (Liu et al., 2010) and thereby elevating $\delta^{18}O_c$ relative to the late Miocene.

### 6.2.3 Imprecision in chronologies and timing of carbonate formation

Our interpretation hinges upon correlating sections across the vast expanse of the Great Plains; however, ages for many Ogallala sections are only poorly constrained, typically via biostratigraphy (Kitts, 1965; Skinner et al., 1977; Thomasson, 1979; Winkler,

1987, 1990; Tedford et al., 2004) or, less frequently, via radiometric dates (Swisher III, 1992; Cepeda and Perkins, 2006; Henry, 2017; Smith et al., 2018). Where biostratigraphy has been used, age resolution is typically limited to the scale of the North American Land Mammal Ages. In many places in eastern New Mexico, the Ogallala Formation is recognized by its clay lithology and its topographic position relative to the laterally extensive caprock to the east (Frye et al., 1978, 1982). Further, the character of Ogallala deposition—large alluvial mega-fans that gradually filled in pre-existing valleys and covered

intervening interfluves combined with its relative thinness—suggest that deposition was often sporadic with frequent and cryptic unconformities associated with many of the paleosols distributed throughout the formation. Consequently, correlations across the entirety of the Ogallala Formation are likely to be imprecise and our treatment of the data likely mixes data from the late middle Miocene and the latest Miocene.

Nevertheless, this imprecision in correlation does not likely affect our conclusions. We note that nearly all of our sections
have low $\delta^{18}O_c$ variability (Fig. 4b; Table 1), suggesting that, throughout deposition of the Ogallala, there was very little change in $\delta^{18}O_p$. Ludvigson et al. (2016), analyzing two cores drilled through the Ogallala Formation in western Kansas, found similarly very low variability in $\delta^{18}O_c$ across the 50-60 m of sampled core material. This low variability, which contrasts with sections located to the west within the North American Cordillera indicate that even large discrepancies in correlated sections do not likely affect our overall conclusion that the south-to-north $\delta^{18}O_p$ gradient has remained similar to today since the late
Miocene.

Further, previously published Neogene $\delta^{18}O_c$ records show almost no change in $\delta^{18}O_c$ during the Miocene (Fox and Koch, 2004; Mix et al., 2013; Chamberlain et al., 2014), again suggesting that imprecision in our correlations are not likely to impact our conclusions. In contrast, modeling studies that have examined changes in $\delta^{18}O_p$ in the Great Plains in the Miocene have suggested that the south-to-north $\delta^{18}O_p$ gradient should increase by several per mille as a consequence of higher $CO_2$
and a shallower equator-to-pole temperature gradient (Feng et al., 2016; Lee, 2019). These studies used boundary conditions thought to approximate the middle Miocene, with atmospheric $CO_2$ of 560 ppm. Disagreement between our results and these model results may arise from a variety of reasons. One such reason may revolve around uncertainty in dating of the Ogallala Formation. While these modeling studies used middle Miocene boundary conditions and 560 ppm $CO_2$, our data likely comes from the late Miocene, following substantial cooling after the peak of Neogene warmth during the middle Miocene (Herbert
et al., 2016). Consequently, the environment in which much of the Ogallala Formation was deposited may differ from that simulated by Feng et al. (2016) and Lee (2019). Alternatively, given the difficulty in simulating the coupling of land-atmosphere over the Great Plains in modern day simulations (Bukovsky et al., 2017; Laguë et al., 2019; Zhou et al., 2021a) and the importance of the land surface in modulating $\delta^{18}O_p$, minor mis-representations in the land surface parameterizations within these models may yield outsize impacts on the simulated $\delta^{18}O_p$ gradient. While a thorough review comparing our results to
simulated Miocene $\delta^{18}O_p$ over the Great Plains is outside the scope of this paper, our new spatially-resolved dataset provides an opportunity to test model predictions of late Miocene climate simulation skill over central North America.

We additionally assume that the authigenic carbonates that we sampled formed in the late Miocene. Several studies (Joeckel et al., 2014; Smith and Platt, 2023) have noted that some caliche units in the northern and central Great Plains may reflect post-depositional case hardening rather than syn-depositional carbonate formation, based upon lateral discontinuities in these
540 units. Given the low variance in our Ogallala samples (typically much less than 1 ‰), we suggest that all of the carbonates in our study formed from the same meteoric water. Further, the $\delta^{13}C$ values of our carbonate samples are relatively low ($\sim -7$ ‰). Given the expansion of $C_4$ plants during the Plio-Pleistocene, the $\delta^{13}C$ values of authigenic carbonates formed in the Plio-Pleistocene is substantially higher ($\sim -2$ ‰) (Fox and Koch, 2003). Thus, the low $\delta^{13}C$ values of our carbonate samples indicates that they formed during the late Miocene under a dominantly $C_3$ grassland environment. The only exception to these

545 low $\delta^{13}C$ values in our sections are from carbonates sampled in eastern New Mexico from outcrops identified as Ogallala by Frye et al. (1982). In many of these sections, there are authigenic carbonate samples with $\delta^{13}C$ values > -5 ‰ and frequently the caprock caliche samples have $\delta^{13}C$ values approaching 0 ‰ (Table S1 in Manser et al. 2023). This suggests that these outcrops may either not be correlative with the Ogallala (Henry, 2017) or that authigenic carbonate formation in these sections occurred substantially later than Ogallala deposition. However, we have no independent age constraints with which to better

constrain the ages of these units, and we therefore include them in our study. We do note that the reconstructed $\delta^{18}O_p$ from these sections are indistinguishable from modern $\delta^{18}O_p$ in eastern New Mexico, further suggesting that the long-term pattern of $\delta^{18}O_p$ in the Great Plains has remained invariant. Thus, the combination of the low $\delta^{13}C$ values combined with the low variance in our $\delta^{18}O_c$ data suggests that our assumption of late Miocene carbonate formation is robust.

## 7 Implications

That atmospheric circulation over the Great Plains has remained relatively constant since the late Miocene provides important context for understanding how Great Plains hydroclimate and environments may respond to higher atmospheric $CO_2$. Further, our data provide insight into the climatic and tectonic controls that may have driven deposition of the Ogallala Formation. In both cases, our data places critical constraints on our understanding of both past environments and the future evolution of climate in the Great Plains.

### 7.1 Response of Great Plains hydroclimate to $CO_2$

Of concern as atmospheric $CO_2$ rises is whether and how the climatological 100th meridian, which demarcates the semi-arid west from the humid east, will shift. The current position of this climatological 100th meridian is partly set today by the boundary between dry westerly air masses that have lost much of their moisture from passage over the North American Cordillera and moist southerly masses transported by the GPLLJ. If dynamical relative shifts in the strength of these two predominant

circulation systems were to occur as $CO_2$ rises, one might expect the boundary between low $\delta^{18}O_p$ and high $\delta^{18}O_p$ to shift. For example, model simulations indicate that the poleward shift of the westerly jet and North Atlantic Subtropical High with warming should enhance GPLLJ moisture transport in the spring, but suppress it in the summer (Zhou et al., 2021a, b). If the overall tendency would be a weaker GPLLJ with warming, we might expect less northward moisture transport and thereby a shift of this $\delta^{18}O_p$ boundary southward and eastward. Instead, we find that the spatial pattern of $\delta^{18}O_p$ has remained unchanged

since the late Miocene, suggesting that the relative strength of these two circulation systems was not substantially different in a warmer, higher $CO_2$ world. Such findings bolster GCM results that, overall, mean wet-season precipitation does not change substantially and that therefore the boundary between the GPLLJ and the westerlies is not overly sensitive to warming.

Though dynamical shifts in circulation may affect hydroclimate over the Great Plains, additional thermodynamic mechanisms have been invoked to explain shifts in the climatological 100th meridian with warming. Because PET will rise faster than

575 P across the Plains, aridity will increase (Seager et al., 2018a, b). However, this will be complicated by how actual ET—and therefore the partitioning of moisture returned to the atmosphere versus to runoff—responds to warming. How ET changes

with warming depends on a variety of factors including plant water use efficiency (Swann et al., 2016; Lemordant et al., 2018) and the temporal distribution of precipitation (Scheff et al., 2022). Our results imply that either (1) the effect of increasing PET relative to P had a small effect on net rainout—and, hence, $\delta^{18}O_p$—in the Miocene, or; (2) decreases in ET efficiency with warming, perhaps driven by more efficient plant water use (Lemordant et al., 2018) and which would reduce $^{18}O$ return to the atmosphere, counteracted the expected increase in $\delta^{18}O_p$ due to increasing PET relative to P.

Though the late Miocene was likely warmer than today with higher than pre-industrial $CO_2$ (Herbert et al., 2016; Mejía et al., 2017; Sosdian et al., 2018; Hönisch et al., 2023), the precise global climate that our new data reflect is uncertain. This uncertainty arises not only because of the chronological uncertainty associated with the Ogallala Formation, but also do to uncertainty in both global temperature and $CO_2$ reconstructions. The late Miocene may have seen global temperatures elevated by 5° higher relative to pre-industrial (Westerhold et al., 2020; Ring et al., 2022) and atmospheric $CO_2$ between 350 and 500 ppm (Tanner et al., 2020; Brown et al., 2022; Hönisch et al., 2023). However, these estimates—particularly for $CO_2$—remain imprecise, and it is therefore difficult to determine just how insensitive Great Plains hydroclimate is to warming and higher $CO_2$. Further, while the land surface plays a critical role in modifying Great Plains hydroclimate, the late Miocene Great Plains was already dominated by grassland ecosystems (Stromberg, 2005). Though $C_4$ plants have a greater water use efficiency than $C_3$ plants (Osborne and Sack, 2012), the invariant $\delta^{18}O_p$ spatial pattern suggests that overall ecosystem water use (*i.e.*, transpiration to the atmosphere) was likely similar to today and overall hydroclimate in the late Miocene resembled hydroclimate today in the Great Plains.

## 7.2   Implications for understanding the origin of the Ogallala Formation

Why much of the Great Plains shifted from a dominantly erosive landscape to a depositional landscape for several million years, before returning to the dominantly erosive landscape of today, has remained a major outstanding question in North American geology (Heller et al., 2003). As the sediments of the Ogallala Formation are sourced in the Rocky Mountains, this question has been intimately linked to what process drove this major late Cenozoic erosional event, producing the alluvial fans and megafans that blanketed the Great Plains in the Late Miocene (Trimble, 1980; Molnar and England, 1990). Paleoaltimetry data have suggested that much of the North American Cordillera has been high since the Eocene (Forest et al., 1995; Mix et al., 2011). If mean elevation has not changed for much of the Cenozoic, then changes in climate—perhaps driven by increasingly variable, orbitally-controlled glacial cycles—may have driven the increase in erosion (Molnar and England, 1990; Gregory and Chase, 1994; Zhang et al., 2001). In contrast, multiple studies that have analyzed paleo-slopes of the Ogallala Formation fluvial sediments conclude that the Ogallala Formation must have been tilted post-deposition due to a long-wavelength uplift (McMillan et al., 2002; Heller et al., 2003), perhaps associated with dynamic topography effects driven by continued subduction of the Farallon plate (Willett et al., 2018).

Our data suggest that hydroclimate was not substantially different in the late Miocene and therefore was unlikely to be the proximal cause of widespread erosion of the Rocky Mountains and deposition of the Ogallala Formation. Instead, we suggest that, because overall hydroclimate—and, in particular, runoff—was similar to today, deposition of the Ogallala Formation must have been driven by long-wavelength tilting of the North American plate that uplifted portions of the Rocky Mountains.

Estimates of the uplift necessary to explain post-depositional tilting are less than 1 km (McMillan et al., 2002; Leonard, 2002). Such uplift was likely not substantial enough to either (1) be detectable in paleo-altimetry datasets or (2) substantially modify the strength of the GPLLJ (Jiang et al., 2007). Uplift might be expected to result in lower $\delta^{18}O_p$ in late Cenozoic basins in the Rocky Mountains due to orographic forcing of precipitation. However, in this region, orographic forcing is limited to GPLLJ storms that reach the Front Range and continued mixing with low $\delta^{18}O_p$ westerly moisture would make it very difficult to discern such a signal. We thus conclude that hydroclimate was not substantially different in the late Miocene in the Great Plains and that long-wavelength tilting must be a necessary component of the formation of the Ogallala Formation.

## 8    Conclusion

Our new spatially resolved late Miocene $\delta^{18}O_c$ data from across the Great Plains reveal that the spatial pattern of reconstructed $\delta^{18}O_p$ is indistinguishable from the spatial pattern of modern meteoric water $\delta^{18}O$. Despite changes in global climate, we suggest that this static $\delta^{18}O$ spatial pattern reflects an atmospheric circulation system over the Great Plains that in the late Miocene was largely identical to today: wintertime westerly air masses, dried by transit over the North American Cordillera, delivered low-$\delta^{18}O$ moisture to the northern Great Plains while southerly moisture, transported by the Great Plains Low-Level Jet, brought high-$\delta^{18}O$ moisture to the southern Great Plains. Thus, on the timescales preserved within the Ogallala Formation, the mixing zone between these two circulation systems was similar to today. Given that these two atmospheric circulation systems are responsible for a nearly 15‰ $\delta^{18}O$ gradient north-to-south along the Great Plains, these results appear to be robust to assumptions regarding temperature, evaporation, and precipitation seasonality changes since the late Miocene and also insensitive to uncertainties in our correlations across the expanse of the Ogallala Formation.

These results suggest that, on geological timescales, large-scale hydroclimate in the Great Plains is relatively insensitive to changes in global temperature and atmospheric $CO_2$. Model projections tend to indicate that mean annual precipitation will not increase substantially as $CO_2$ rises (Bukovsky et al., 2017; Zhou et al., 2021a), and our results support this contention, in that a static $\delta^{18}O$ spatial pattern suggests that net rainout in the late Miocene across the Great Plains was likely not substantially different than today. In contrast, our results do not support projections that aridity will increase over the Plains (Seager et al., 2018a). While increases in PET are robust in models (Scheff et al., 2017), our results suggest that countervailing decreases in actual ET efficiency likely offset any change in PET, yielding an approximately constant spatial pattern of P/ET. Further, our results indicate large-scale changes in climate (Zhang et al., 2001) are likely not the cause of the widespread erosional event responsible for deposition of Ogallala sediments. Instead, the fact that Great Plains hydroclimate in the late Miocene was similar to today supports the notion that long-wavelength uplift (Moucha et al., 2009; Karlstrom et al., 2011; Willett et al., 2018) shifted the Great Plains from a dominantly erosive landscape to an aggradational one.

The Great Plains lie at a unique climatic and geologic intersection, and this confluence has inspired more than a century of work to understand the relationship between climate, geology, and ecosystems over geologic time in the Plains (Powell, 1879, 1890; Seni, 1980; Fox and Koch, 2003; Jiang et al., 2007; Seager et al., 2018b; Willett et al., 2018). However, the sharp geologic and climatic gradients that characterize this landscape complicate efforts to understand past changes. Nevertheless,

our results suggest that spatially resolved datasets provide a powerful means to constrain the position and the shape of these
645 gradients in the past. Additional work that utilizes newly developed proxy systems and/or pairs well-established proxies with
our continental-scale data, will provide an even sharper picture of the climatic and geologic forces that have shaped this
remarkable landscape.

## Appendix

**Table 2.** Input parameters for the reactive transport model. All inputs retrieved from NARR data except $\omega$, which is set to the global average
of 2.6 (Greve et al. (2015)), and the transpired fraction of ET, which is set to 0.64 (Good et al. (2015)).

| Parameter | GPLLJ Trajectory | Westerly Trajectory |
|---|---|---|
| **Temperature (MAT)** | 295 K | 287 K |
| **Potential ET (PET)** | 2150 mm yr-1 | 1960 mm yr-1 |
| **Wind speed (U)** | 2 m s-1 | 3.2 m s-1 |
| **Precipitable water (W)** | 33.6 kg m-2 | 16.2 kg m-2 |
| **Relative humidity (RH)** | 80% | 74% |
| **Dryness index (DI)** | 2.2 | 3.4 |
| $\omega$ | 2.6 | 2.6 |
| **Transpired fraction of ET (T/ET)** | 0.64 | 0.64 |

*Code availability.* The R code for the vapor transport model has been published as a supplement to Kukla et al. (2019)

.

*Data availability.* The data sets used in this study are listed below.
- Compiled, published $\delta^{18}O$ data (Appendix Table 3, accessed via the PATCH Lab (https://geocentroid.shinyapps.io/PATCH-Lab/) (Kukla
et al., 2022a)
- Modern water isotope data (Waterisotopes Database, 2019)
- North American Regional Reanalysis (Mesinger et al., 2006)
- Table S1, available at the data repository Dryad (Manser et al., 2023)

*Author contributions.* JKCR conceived the project idea, acquired funding, and provided overall supervision; LM conducted the investigation,
curated the data, conducted the formal analysis, and prepared the original draft of this manuscript; TK provided resources and supervised the
use of the vapor transport model; all authors contributed to the review and editing of the manuscript.

**Table 3.** Site-averages for previously published data, accessed via the PATCH Lab (Kukla et al. (2022a)). Values of $\delta^{13}C$ reported in ‰ relative to VPDB. Values of $\delta^{18}O$ reported in ‰ relative to VSMOW. No standard deviation or range reported for sites that only include one sample.

| Lat (°N) | Lon (°E) | $\delta^{13}C$ | $\delta^{13}C\ 1\sigma$ | $\delta^{18}O_c$ | $\delta^{18}O_p$ | $\delta^{18}O\ 1\sigma$ | $\delta^{18}O$ range | Age (Ma) | MAT (°C) | Reference |
|---|---|---|---|---|---|---|---|---|---|---|
| 28.29 | -97.97 | -7.29 | 1.29 | 25.75 | -3.09 | 0.74 | 2.65 | 8.50 | 22.81 | Godfrey et al. 2018 |
| 28.34 | -97.97 | -5.49 | 0.67 | 25.99 | -2.86 | 0.26 | 0.87 | 8.50 | 22.82 | Godfrey et al. 2018 |
| 28.50 | -98.10 | -7.27 | 0.81 | 26.39 | -2.44 | 0.30 | 0.70 | 8.00 | 22.89 | Godfrey et al. 2018 |
| 28.65 | -97.38 | -9.62 | 0.10 | 27.02 | -2.01 | 0.11 | 0.32 | 8.50 | 22.00 | Godfrey et al. 2018 |
| 28.80 | -97.80 | -4.65 | 1.89 | 26.34 | -2.65 | 0.66 | 1.90 | 8.00 | 22.14 | Godfrey et al. 2018 |
| 33.41 | -101.56 | -6.51 | 0.68 | 26.41 | -3.64 | 0.49 | 1.70 | 6.80 | 17.39 | Fox and Koch 2003 |
| 33.50 | -101.60 | -5.91 | 0.36 | 26.58 | -3.52 | 0.34 | 1.90 | 13.05 | 17.14 | Fox and Koch 2003 |
| 34.90 | -103.10 | -7.20 | 0.33 | 25.81 | -4.69 | 0.23 | 0.80 | 6.40 | 15.37 | Fox and Koch 2003 |
| 35.70 | -100.50 | -6.92 | 0.89 | 26.16 | -4.24 | 0.19 | 0.60 | 6.65 | 15.82 | Fox and Koch 2003 |
| 36.10 | -100.00 | -7.37 | 0.32 | 25.83 | -4.69 | 0.47 | 0.90 | 9.60 | 15.29 | Fox and Koch 2003 |
| 36.18 | -100.00 | -7.41 | 0.91 | 25.92 | -4.62 | 0.42 | 1.10 | 8.75 | 15.24 | Fox and Koch 2003 |
| 39.40 | -100.10 | -6.99 | 0.44 | 23.53 | -7.60 | 0.57 | 1.60 | 9.55 | 12.72 | Fox and Koch 2003 |
| 41.20 | -103.70 | -5.57 | 0.78 | 21.31 | -10.36 | 0.43 | 1.50 | 7.25 | 10.47 | Fox and Koch 2003 |
| 41.24 | -101.81 | -7.04 | 0.24 | 19.32 | -12.24 | 0.95 | 2.80 | 7.25 | 10.92 | Fox and Koch 2003 |
| 41.30 | -102.40 | -6.63 | 0.44 | 19.30 | -12.29 | 0.85 | 3.70 | 8.65 | 10.82 | Fox and Koch 2003 |
| 41.47 | -103.07 | -6.60 | 0.14 | 17.73 | -13.90 | 0.15 | 0.21 | 7.00 | 10.65 | Fan et al. 2014 |
| 41.50 | -103.10 | -6.79 | 0.45 | 18.64 | -12.99 | 1.05 | 4.40 | 9.10 | 10.63 | Fox and Koch 2003 |
| 41.60 | -102.78 | -7.00 | NA | 18.34 | -13.32 | NA | 0.00 | 9.00 | 10.53 | Fan et al. 2014 |
| 42.37 | -107.06 | -6.50 | NA | 15.35 | -17.20 | NA | 0.00 | 10.50 | 6.93 | Fan et al. 2014 |
| 42.40 | -98.20 | -7.00 | 0.28 | 22.90 | -8.95 | 0.14 | 0.20 | 12.69 | 9.74 | Fox and Koch 2003 |
| 42.40 | -98.20 | -8.36 | 0.91 | 24.84 | -7.01 | 0.93 | 2.60 | 13.70 | 9.74 | Fox and Koch 2003 |
| 42.40 | -103.22 | -7.18 | 0.50 | 22.56 | -9.25 | 0.92 | 2.90 | 17.50 | 9.89 | Fox and Koch 2003 |
| 42.49 | -103.85 | -7.60 | 0.14 | 18.29 | -13.59 | 0.37 | 0.52 | 18.50 | 9.63 | Fan et al. 2014 |
| 42.58 | -107.19 | -6.10 | 0.17 | 13.57 | -18.92 | 1.64 | 2.89 | 14.17 | 7.19 | Fan et al. 2014 |
| 42.72 | -108.19 | -3.27 | 2.41 | 17.06 | -15.79 | 1.94 | 5.73 | 13.02 | 5.72 | Chamberlain et al. 2012 |
| 42.80 | -100.00 | -6.49 | 1.16 | 21.67 | -10.16 | 1.61 | 3.80 | 14.30 | 9.80 | Fox and Koch 2003 |

*Competing interests.*  The authors declare that they have no conflict of interest.

*Acknowledgements.*  This study benefited from discussions with Sean Willett regarding the Ogallala Formation and landscape evolution. We thank Sean for first piquing our interest in this unique and still mysterious paleo-landscape. We thank Madalina Jaggi, Ulrich Treffert, Niklaus Loeffler, and Emilija Krsnik for assistance in the laboratory, Katharina Methner for field assistance, and Dan Koning and Frank Pazzaglia for assistance with finding key outcrops in New Mexico. We also benefited from discussions with Vincenzo Picotti, Giuditta Fellin, and

Derek Sjostrom regarding Ogallala paleosol and sandstone petrology. We thank Rusty Winn of RE Janes Gravel Co (Lubbock, TX, USA) for facilitating our access and guiding us to remarkable Ogallala outcrops in the southern Great Plains. We also thank Phillip Osborne for permission to sample on his property, and the Wildcat Bluff Nature Center for permission to sample within their preserve. Samples from Lake McConaughy State Recreation Area were collected under a permit issued by the Central Nebraska Public Power & Irrigation District, and samples collected at Palo Duro and Caprock Canyons State Parks in Texas were collected under permit 2018-R5-08. Lastly, we thank an

anonymous review, Dr. David Fox, and Dr. Jon Smith for reviews that substantially improved this manuscript. This study was funded by an ETH SEED Grant, ETH and Alexander von Humboldt postdoctoral fellowships, and NSF grant EAR-2202916 to Rugenstein.

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
