# Peer review of "Stable isotope evidence for long-term stability of large-scale hydroclimate in the Neogene North American Great Plains"

_EGUsphere, 2023_

## Editor Comment (EC1)

Comments by Jon Smith

The manuscript is well written, well organized and would be of interest to many geoscientist, but If the authors are not able to address these concerns, I think it will be very skeptically received.
While I did not get a chance to fully review the manuscript, I did identify some very concerning issues regarding the vintage and accuracy of the geologic background information and the field methods and assumptions with respect to how rock samples were selected in the field.  Despite my tardy reply, I list my concerns below.

1. Line 50 - References to the overall and specific geology composition of the Ogalla are very dated, which isn't necessarily a problem except that it not consulting more recent studies likely led to the additional lithostratigraphic concerns listed below.  The authors should carefully read the more recent papers specifically addressing the lithofacies and calcrete stratigraphy of the Ogallala such as:
   a. Joeckel, R.M., Wooden Jr, S.R., Korus, J.T., and Garbisch, J.O., 2014, Architecture, heterogeneity, and origin of late Miocene fluvial deposits hosting the most important aquifer in the Great Plains, USA: Sedimentary Geology, v. 311, p. 75-95.
   b. Smith, J.J., and Platt, B.F., 2023, Reconstructing late Miocene depositional environments in the central High Plains, USA: Lithofacies and architectural elements of the Ogallala Formation: Sedimentary Geology, v. 443, p. 106303.
   c. Korus, J., and Joeckel, R.M., 2023, Telescopic Megafans on the High Plains, USA Were Signal Buffers in a Major Source-To-Sink System: The Sedimentary Record, v. 21.

2. Line 52 - The concept of the Ogallala "cap rock" referring to a regionally persistent and ledge-forming terminal petrocalcic horizon is not supported by more recent stratigraphic studies (Swineford et al. 1958; Diffendal 1982; Gustavson and Winkler 1988; and Joeckel et al. 2014). Instead, these studies show that carbonate-cemented paleosols and petrocalcic horizons are present in numerous stratigraphic positions in Ogallala deposits.  I understand that there is a very prominent calcrete at the contact between the Ogallala and the Blackwater Draw Formation in TX, but this should not be interpreted to represent a regionally persistent marker bed throughout the expanse of the Ogallala as was previously assumed (prior to the 2000s).  See References above and Ludvigson et al. (2009), Review of the stratigraphy of the Ogallala Formation and revision of Neogene ("Tertiary") nomenclature in Kansas.

3. Line 68 - see earlier comment, while I agree Ogallala exposures are typically well indurated by carbonate, I would hesitate to refer to this characterstic as "its caprock", as its not a single bed.

4. Figure 1- giving the circles and squares different colors might help to distinguish them a bit better.

5. Line 175 – Some very interesting papers have recently been published on just this topic, see Korus, J., and Joeckel, R.M., 2023, Telescopic Megafans on the High Plains, USA Were Signal Buffers in a Major Source-To-Sink System: The Sedimentary Record, v. 21.

6. Line 182 – Additional publications with specific volcanic age data from Ogallala ash bed should be cited:
   a. Swisher, C. C. III, 1992, $^{40}Ar/^{39}Ar$ dating and its application to the calibration of the North American land mammal ages [Ph.D. thesis]: Berkeley, University of California, 239 p.
   b. Smith, J.J., Turner, E., Moller, A., Joeckel, R.M., and Otto, R.E., 2018, First U-Pb zircon ages for late Miocene Ashfall Konservat-Lagerstatte and Grove Lake ashes from eastern Great Plains, USA: Plos One, v. 13.

7. Line 192 - This may be the case in some areas, but it is not a consistant feature. Calcretes are present in many Ogallala outcrops regardless of stratigraphic position as it is most like a result of exposure case hardening. See Joeckel et al. 2014 and Smith and Platt 2023 for more on modern interpretation of "cap rock".

8. Line 205 – "on the assumption that the caprock formed simultaneously across the Great Plains." – we know this is incorrect. See references above.

9. Line 211 – See Smith and Platt (2023) for more on unconformities and the thorny issue of Ogallala calcretes…

10. Line 226 - The authors need to provide more information on their samples and sampling methods. There are many carbonate morphologies in the Ogallala, and its becoming increasingly clear most are not coincident with paleosol formation. The pics in Fig. 2 helps, but is also concerning. Carbonate nodules and burrows may be authogenic, but I have some reservations about the pictured root casts and am very skeptical of the "cap rock" calcrete. We strongly suspect that many of these calcretes and calcrete morphologies are primarily carbonate precipitation due to case hardening of the exposed surface and not syndepositional. This is a vitally important issue because this may be the primary reason you are getting a consistently modern signal from your d18O; you may be sampling carbonates that precipitated essentially in response to recent exposure and under essentially modern conditions. I'm not stating that is the case, but its impossible for me to tell without being more specific in how and what you sampled.

11. Line 317 – "this year"…. What year? 2016? Or an average of 1980-2016?

12. Line 421 – "also imprecision in the chronologies of the sections we sampled"… Not just the sections, but the sampled material itself, as in assuming the carbonate is in some respect syndepositional with the host sediments. I would be curious to see inter-area sampling differences. For example what is the variance in d18O between the 19 samples from the BV location? Are there patterns with respect to sample type (nodules that appear pedogenic vs calcrete vs rhizoliths)?

Again, I apologize for not completing my review. I have few concerns about the results of the geochemical analyses. The methodology and output is well communicated, and I would not be surprised by their findings in the slightest; in fact they align very well with my most recent publication using paleosols and trace fossils to interpret climate conditions (Platt, B.F., and Smith, J.J., 2023. Late Miocene paleoecology and paleoclimate in the central High Plains of North America reconstructed from paleopedological, ichnological, and stable isotope analyses of the Ogallala Formation in western Kansas, USA. Evolving Earth, https://doi.org/10.1016/j.eve.2023.100019.) But frankly, I am extremely skeptical of their interpretations and conclusions due to the lack of communicating exactly what was sampled, how or why they suspect the sampled carbonate is ancient, and the authors out dated understanding of the regional geology. I was not able to complete my review, so I will not make a final recommendation.

---

## Author Comment (AC1)

Comments by the reviewer are *italicized*; our responses are in indented, normal font.

*Manser et al.*

*Stable isotope evidence for long-term stability of large-scale hydroclimate in the Neogene North American Great Plains*

*EGUSphere*

*Comments of David Fox*

*Summary*

*The paper presents new d18O data from Miocene pedogenic carbonates from the Ogallala Group/Formation in the Great Plains region of the USA and, in conjunction with compiled published data, analyses the spatial pattern in relation to patterns predicted from climate data and moisture transport model. The paper is well-conceived and clearly written and is mostly thorough in its treatment of the data, the geological context, and the modern and Miocene climate system and implications of the data for our understanding of Miocene hydroclimate in central North America. I have a few mostly minor comments below and think this paper is ready suitable for publication after minor corrections and additions.*

> We thank Dr. Fox for the thorough review and thoughtful comments. Addressing these concerns has certainly made the paper better.

*General comments.*

*1. The lithostratigraphic terminology for the Ogallala is a bit complicated across the region. In some states (e.g., Nebraska) it is a Group with constituent formations and in others (e.g., Kansas) it is an undifferentiated Formation. I think Tedford et al., 2004 (full reference below) is the most authoritative source for the current terminology. I suggest that you address this complexity and explain your use and be consistent throughout (which you are now, but by treating it as a Formation, which is not correct everywhere in the study region).*

> We have now modified section 3 to incorporate a more nuanced discussion of Ogallala lithostratigraphic terminology and to emphasize differences in the north-to-south extent of the Ogallala. We also continue to use Formation when referring to the Ogallala, but explain that, in the northern Great Plains, the Ogallala is considered a Group.

*2. Include all of the data in a supplemental with this paper, both the previously published and your new data, rather than really solely on Dryad for the new data. I could not access the new data on Dryad, though maybe it is not yet accessible or maybe it is pilot error? Regardless, I think including all of the underlying data in a supplemental with this paper is critical even with the data available via Dryad as well.*

We apologize that the data was not available and admit that this was due to the fact that we mis-understood the directions on Dryad. The data should now be available at the following doi: 10.5061/dryad.5hqbzkhc5; however, if it is not yet available, please use this link to access it: https://datadryad.org/stash/share/sY5SXxVH-Hfp104CZ4m-grrcnOh-RAdwdbCNrF1Avcg. Climate of the Past appears to strongly prefer that datasets are presented not as supplemental material.

*3. Address explicitly the age assignments and uncertainties. You discuss age uncertainty as a factor in comparisons to published model results, but you do not discuss in any detail the uncertainties in your age assignments beyond listing maximum and minimum ages in Table 1. What is the basis for the assignment of each section and how did you treat the uncertainty? Given that you do not examine the data as time series, this is not so much an issue with your interpretations as a matter of completeness.*

We now explicitly address how we assign age constraints to our samples in lines 221-236. In short, we rely upon published studies that have studied these sections in more detail. In some cases, there is dateable material that provides a constraint; in others, the studied section is well-correlated to a dated section that provides constraints. And in still other sections, there is only a lithologic correlation to the Ogallala and no dateable material. We therefore adopt the broadest possible age constraints for samples from these types of sections.

*4. Do you plan a separate paper for the carbon isotope data? If so, then perhaps including the new carbon isotope data from carbonates here is not necessary. However, if you do not plan a separate paper on the carbon data, then I suggest that you include at least the new carbon data (and probably all of the published carbon data as well) in the supplemental table of data and add a few sentences summarizing how the new carbon data compare to the published data. I recognize the carbon data are not the focus of this paper, and you can qualify some brief comments by saying a detailed discussion is beyond the scope of this paper, but I worry that the carbon data will be orphaned and lost if you do not plan to publish them and do not include them here. The published Miocene data (my papers that you cite and Lukens and Fox (2022. Palaeogeography, Palaeoclimatology, Palaeoecology 586. doi.org/10.1016/j.palaeo.2021.110760) have a strong central tendency (-7.2 to -6.8‰ V-PDB) and low variance, so a simple comparison of your mean and s.d. to the published data would suffice.*

We are working on a separate manuscript that will describe the $\delta^{13}C$ data in detail. The $\delta^{13}C$ values are listed in Table S1 as well (though we realize that this dataset was mistakenly withheld from the reviewers). However, the $\delta^{13}C$ data do add to our interpretation (see response to Reviewer Dr. Smith) and, in general, are useful to include in such a paper and we therefore now append the $\delta^{13}C$ data to Table 1 (main text). Because we are working on a separate paper, we do not discuss the patterns in the $\delta^{13}C$ data in detail.

All of our new Ogallala data (*i.e.*, excluding data from the Santa Fe Group) has a mean $\delta^{13}C$ of -6.14 ‰ ± 1.39 (1σ). However, when we exclude data from sites primarily in

New Mexico that may be younger than late Miocene (see discussion in response to Reviewer Dr. Smith) and that Frye et al. (1982) correlated to the Ogallala based only on lithologic or geomorphic relationships, then the mean $\delta^{13}C$ for our data is -6.69 ‰ ± 0.88 (1σ). Similarly, the mean $\delta^{13}C$ of the compiled data we present in our paper (mostly, but not exclusively compiled from Fox and Koch (2003)) is -6.74 % ± 0.83 (1σ). We view this similarity as sufficient to support our contention that—outside of the Frye et al. (1982) sites—all of our samples record late Miocene climate.

We now include these summary statistics of our $\delta^{13}C$ data in lines 351-362.

*Specific comments (indexed by line number)*

*36. It might be worth noting the influence of longer timescale climate fluctuations here, particularly ENSO. As I understand, the onset of the Dust Bowl coincided with a particularly severe El Nino event, and the little Dust Bowl in the 1950s also coincided with a string El Nino. These longer frequencies in the climate system are obviously not the focus of your paper, but they are relevant in the intro it seems.*

We have now included brief mentions of long-term climatic oscillations that have been invoked to explain extreme droughts and floods in the region.

*39. Is Powell's work relevant beyond being antecedent? He is a somewhat complicated figure historically, perhaps less so than others, but the mention of him does not do much work here. He is traditionally treated as somewhat of a founder and hero in North American geology and geomorphology, but I am not sure everyone in North America views him so positively. That said, I don't think your treatment is problematic.*

As you point out, this mention of Powell does not add to the manuscript, so we have eliminated this phrase.

*48. Check on the regional nomenclature for the Ogallala. It is a Group in Nebraska with multiple constituent Formations, but I think used as an undifferentiated Formation elsewhere in the region, certainly in Kansas (see Ludvigson et al., 2009). My sense is that in Texas, different authors use it as either a formation or a group, but I am not sure which is currently formally correct. You should point this out and establish here a terminology that you will use throughout for the lithostratigraphic unit. For example, you could use Ogallala for the unit and specify "Ogallala Aquifer" when refering to the aquifer. I think Tedford et al. (2004) (see note for line 182) is the most definitive authority on the regional terminology.*

We have now clarified the terminology that we will be using in this paper in section 3 (lines 188-191). We also have re-worked section 3 to provide a more thorough background on the regional differences in the Ogallala Formation (*i.e.*, that it is considered a group in the northern Great Plains, but an undifferentiated formation in the southern Great Plains).

*51. To this list of references, thanks to my slow review, you can add Korus and Joeckel, 2023. Telescopic Megafans on the High Plains, USA Were Signal Buffers in a Major Source-To-Sink System. The Sedimentary Record 21. https://doi.org/10.2110/001c.89096.*

> Now included.

*52. The Ogallala is not capped by a single, laterally continuous caprock. This idea was prominent in the early literature on the stratigraphy in the region, but is not correct.*

> We have modified the wording here to denote that this "caprock" is really only present in the southern Great Plains and that it appears in places there, but we do not mean to imply that it is regional extensive.

*54. In Nebraska, the Ogallala also lies on top of formations of the Arikaree Group.*

> Noted and included now in the manuscript.

*148. Should this be "between the land surface and the atmosphere"?*

> Yes, now fixed.

*161. I am not sure either of these are the best citations for the orographic effects on precipitation amount and d18O (e.g., Rozanski et al., 1993).*

> We have now included additional references here.

*182. You should cite Tedford et al. (2004) here as the most recent detailed synthesis of the mammalian biostratigraphy in the region for the study interval, and it includes more or less all of the reliably dated ashes to date. Tedford, R.H., Albright III, L.B., Barnosky, A.D., Ferrusquia-Villafranca, I., Hunt Jr., R. M., Storer, J.E., Swisher III, C.C., Voorhies, M.R., Webb, S.D., Whistler, D.P., 2004. Mammalian Biochronology of the Arikareean through Hemphillian Interval (Late Oligocene through Early Pliocene Epochs): Late Cretaceous and Cenozoic Mammals of North America: Biostratigraphy and Geochronology. Columbia University Press, New York, pp. 169–231.*

> We have now included this citation here as well as in additional locations in the manuscript.

*183. See my earlier comment about lithostratigraphic nomenclature.*

> We now use consistent terminology throughout to describe the Ogallala Formation.

*186. The citations here could include Tedford et al. (2004), but also more primary literature on each unit.*

> We have now added additional references here as well as Tedford et al. (2004).

*189. "in Texas" needs to be moved as the Blackwater Draw Formation is only in Texas and only overlies the Ogallala in places there. You need to be clear and specific about this here as the Ogallala is overlain by high energy deposits in places elsewhere (i.e., the Stump Arroyo Mbr of the Crooked Creek Fm in SW Kansas and the Broadwater Fm in W Nebraska).*

> We have now modified these sentences to better describe the regional variability in the overlying sedimentary units.

*192. The Ogallala includes multiple stratigraphically distinct cap rocks or mortar beds and not one regionally extensive or continuous one and not only one at the top of the section.*

> We now note that this prominent caprock exists only in the southern Great Plains that separates the Ogallala from the overlying Blackwater Draw Fm.

*203. As suggested before, you should clarify the stratigraphic nomenclature earlier and make sure it is complete and accurate.*

*207. This is true almost everywhere and it is well documented that the caprock is not a single unit stratigraphically.*

> We have now clarified that, when we discuss the caprock, we are primarily referring to the "caprock" in the southern High Plains and do not mean to imply that it is a single bed nor formed at a single time in Earth history. We hope this language now clarifies that.

*219. Could add Tedford, 1981. Mammalian biochronology of the late Cenozoic basins of New Mexico. Geological Society of America Bulletin 92: 1008-1022.*

> Added.

*228. The references for the age assignments for each section should be given in Table 1 so that readers can evaluate the age assignments on their own. You need to state here how you assign a specific age to each section and/or sample given that the sections have age ranges. Are all samples in a section given the same age? Do you assume a sedimentation rate and assign ages in stratigraphic sequence, and, if so, how do you calculate sedimentation rate?*

> We now clarify in this paragraph (lines 221-236) how we assign ages to the samples. We have now updated Table 1 to also include the references that provide the age constraints. We do not, however, that our analysis strategy is not particularly sensitive to the precise ages of individual samples in our sections.

*235. Be explicit about how the standards were used…Which was used to correct to the V-SMOW scale and which were used for runtime QA/QC? This needs to be reported clearly. What is the analytical precision and how was that determined?*

> This has now been spelled out in detail in the methods. We use the IUPAC-recommended equations in Brand et al. (2014) to convert our data from VPDB to VSMOW. No

standards are used for runtime QA/QC due to the consistently lower than 0.1‰ reproducibility of the standards, and this is standard operating protocol at the ETH Stable Isotope Laboratory (S. Bernasconi, personal communication, 2024).

*295. Here and elsewhere, what is the basis for the order of citations? It does not seem to be alphabetical nor chronological.*

Climate of the Past does not appear to have a preference for the ordering of citations.

*331. The table of published data should include the original published C and O values and your calculated paleoprecipitation values.*

We now include the original published carbonate $\delta^{13}C$ and $\delta^{18}O$ values, with the $\delta^{18}O$ values converted to VSMOW.

*332. I see no reason not to include the full data table as a supplemental to this paper also so that the data are with the interpretation. I cannot access the data using the doi nor by searching on Dryad, though perhaps the data are not yet posted or accessible?*

It appears that Climate of the Past strongly discourages supplements of this nature. We have now ensured that this data is publicly available via Dryad (please use this link if the doi remains unavailable: https://datadryad.org/stash/share/sY5SXxVH-Hfp104CZ4m-grrcnOh-RAdwdbCNrF1Avcg).

*434. Higher is probably a better word choice here than greater.*

Fixed.

*469. Tedford et al., 2004*

Added

*537. One factor you do not discuss is the difference in water use efficiencies and evapotraspiration fluxes of woody vegetation vs. C3 grasses vs. C4 grasses. This is embedded in your consideration of land surface characteristics, but there is literature on this could be of use. The phytolith data suggest that grasslands were present throughout the Miocene, and the carbon isotope data have been interpreted as indicating a constant amount of C4 grasses throughout all of or almost all of the Miocene, so these patterns are consistent with your lack of a spatial signal in the d18O data.*

Thanks for pointing this out. We now include a few sentences (lines 590-592) discussing this point.

*545. data have*

Fixed.

**References cited in response to reviewer**

Brand, W.A., Coplen, T.B., Vogl, J., Rosner, M., Prohaska, T., 2014. Assessment of international reference materials for isotope-ratio analysis (IUPAC technical report). Pure Appl. Chem. 86, 425–467. https://doi.org/10.1515/pac-2013-1023

Fox, D.L., Koch, P.L., 2003. Tertiary history of C4 biomass in the Great Plains, USA. Geology 31, 809–812. https://doi.org/10.1130/G19580.1

Frye, J.C., Leonard, A.B., Glass, H.D., 1982. Western extent of Ogallala Formation in New Mexico. New Mex. Bur. Mines Miner. Resour. Circ. 175, 41.

Kim, S., O'Neil, J., 1997. Equilibrium and nonequilibrium oxygen isotope effects in synthetic carbonates. Geochim. Cosmochim. Acta 61, 3461–3475.

Tedford, R.H., Albright, L.B., Barnosky, A.D., Ferrusquia-Villafranca, I., Hunt, R.M., Storer, J.E., Swisher, C.C., Voorhies, M.R., Webb, S.D., Whistler, D.P., 2004. Mammalian Biochronology of the Arikareean Through Hemphillian Interval (Late Oligocene Through Early Pliocene Epochs), in: Woodburne, M.O. (Ed.), Late Cretaceous and Cenozoic Mammals of North America. Columbia University Press, New York, NY, USA, pp. 169–231. https://doi.org/10.7312/wood13040-008

---

## Author Comment (AC2)

Comments by the reviewer are *italicized*; our responses are in indented, normal font.

*Review by anonymous reviewer 1:*

*This paper presents an analysis of oxygen isotopes in modern water isotopes and ancient paleosols from the Great Plains. The authors apply a vapor transport model to see whether this simple 1-D model can explain observed gradients of isotopes (the model does relatively poorly), and also analyze whether these gradients have changed in the past. The abstract and paper text are well written, and it is overall appropriate for the journal. Some conceptual considerations:*

> We thank the reviewer for their thoughtful comments.

*Table 1 is helpful, but it would be helpful to have a map figure where sites are color-coded by their lower age and/or upper age.*

> All of the samples presented are Miocene in age. In this study, since we are not comparing the sections across different stages of the Miocene (and to keep the figure readable), we stay with the current figure (though see changes in response to Reviewer Dr. Smith). All ages are available in Table 1.

*The mismatch between the reactive transport model and observations is quite dramatic, and therefore I would like a greater discussion of the sources of mismatch between the reactive transport model and the data. Rainout is definitely a factor that can affect isotopic gradients, but what about storm statistics and the changing location of certain types of storms (see papers below)?*

*Sun, C., Shanahan, T.M., DiNezio, P.N., McKay, N.P. and Roy, P.D., 2021. Great Plains storm intensity since the last glacial controlled by spring surface warming. Nature Geoscience, 14(12), pp.912-917*

*Maupin, C.R., Roark, E.B., Thirumalai, K., Shen, C.C., Schumacher, C., Van Kampen-Lewis, S., Housson, A.L., McChesney, C.L., Baykara, O., Yu, T.L. and White IV, K., 2021. Abrupt Southern Great Plains thunderstorm shifts linked to glacial climate variability. Nature Geoscience, 14(6), pp.396-401.*

> Here, rainout and the effect of changing storm statistics/storm locations on $\delta^{18}O$ are simply different sides of the same coin – the climatological pattern of rainout is the time integration of many storm events, each characterized by its own rainout pattern. We feel that extending our interpretation to infer specific storm-scale features (rather than the more general diagnosis of rainout) would go beyond what our data reasonably permit.

> In any case, comparisons with the Last Glacial Maximum are potentially not useful in understanding geologic shifts in $\delta^{18}O$ for a couple of reasons. First, both papers demonstrate that changes in the ice sheet extent (particularly the Laurentide ice sheet) affect the intensity of the Great Plains Low-Level Jet (GPLLJ). In turn, changes in ice sheet extent between now and the late Miocene do not affect GPLLJ intensity. Second,

Sun et al. (2019) show that springtime insolation—modulated by precession—impacts GPLLJ intensity via changes in the zonal pressure gradient. Precession is a large unevenly distributed forcing change on the Earth system; in contrast, changes between the modern and the late Miocene are more likely driven by $CO_2$, which provides a more globally uniform forcing and is therefore not directly comparable to orbital changes. Plus, our Miocene samples likely integrate signals across orbital timescales, given the timescale of formation of authigenic carbonates (Berner, 1968). Thus, we do not expect to see orbitally driven variations in $\delta^{18}O$ in our dataset.

*Another thing I noticed is that the paleosols in this study extend to roughly 40-45 N. If you look at the climatology over the modern GPLLJ, in the modern climatology the jet counts (e.g. calculated on daily data) extend to roughly 45 N:*

*Helfand, H.M. and Schubert, S.D., 1995. Climatology of the simulated Great Plains low-level jet and its contribution to the continental moisture budget of the United States. Journal of Climate, 8(4), pp.784-806.*

*Would you actually need sites that are even farther north to detect poleward extensions/expansions of the GPLLJ, especially in past warm climates? It seems that the latitudinal range of samples in this study would be most appropriate for detecting contractions of the jet's intensity or northward extent? Are there past changes that could not be detected by the current dataset? However, I do agree with the overall conclusion about air masses, since if North American topography was high in the Miocene, we would expect a general pattern of mixing between the low level flow and midlatitude air masses.*

While having additional spatial data will always help improve interpretations of spatial patterns in $\delta^{18}O$, we are not aware of any data to the north of the Ogallala Formation during our study time frame. With that said, the core of the jet, according to Helfand and Schubert (1995), lies within the middle of our dataset (between ~33° and 43° N) allowing us to resolve changes in GPLLJ intensity.

*Dynamically, future changes in the GPLLJ have been linked to changes in the position of the North Atlantic Subtropical High, and this literature should be discussed in the manuscript (another paper by this team is already cited in the MS). I may be wrong, but Zhou et al are specifically references the westerly jet position over the Atlantic and its relationship to future changes in the GPLLJ, not necessarily the upstream jet over the west coast*

*Zhou, W., Leung, L.R., Song, F. and Lu, J., 2021. Future changes in the Great Plains low-level jet governed by seasonally dependent pattern changes in the North Atlantic subtropical high. Geophysical Research Letters, 48(4), p.e2020GL090356.*

We thank the reviewer for this reference and have included this reference as well as a short discussion of changes in the North Atlantic Subtropical High and links to the GPLLJ (lines 566-567).

**References cited in review**:

Berner, R.A., 1968. Rate of concretion growth. Geochim. Cosmochim. Acta 32, 477–483. https://doi.org/10.1016/0016-7037(68)90040-9

Helfand, H.M., Schubert, S.D., 1995. Climatology of the simulated Great Plains low-level jet and its contribution to the continental moisture budget of the United States. J. Clim. https://doi.org/10.1175/1520-0442(1995)008<0784:COTSGP>2.0.CO;2

Kukla, T., Winnick, M.J., Laguë, M.M., Xia, Z., 2023. The Zonal Patterns in Late Quaternary Tropical South American Precipitation. Paleoceanogr. Paleoclimatology 38, 1–21. https://doi.org/10.1029/2022PA004498

Moore, M., Kuang, Z., Blossey, P.N., 2014. A moisture budget perspective of the amount effect. Geophys. Res. Lett. 41, 1329–1335. https://doi.org/10.1002/2013GL058302.Abstract

Sun, C., Shanahan, T.M., Partin, J., 2019. Controls on the Isotopic Composition of Precipitation in the South-Central United States. J. Geophys. Res. Atmos. 124, 8320–8335. https://doi.org/10.1029/2018JD029306

---

## Author Comment (AC3)

Comments by the reviewer are *italicized*; our responses are in indented, normal font.

*Comments by Jon Smith*

*The manuscript is well written, well organized and would be of interest to many geoscientist, but If the authors are not able to address these concerns, I think it will be very skeptically received.*
*While I did not get a chance to fully review the manuscript, I did identify some very concerning issues regarding the vintage and accuracy of the geologic background information and the field methods and assumptions with respect to how rock samples were selected in the field. Despite my tardy reply, I list my concerns below.*

> We thank Dr. Smith for his thorough and helpful review and, in particular, the references to newer literature. Dr. Smith's comments have helped improve our overall description of the stratigraphy and we also address in detail (both here in the review and also in the manuscript) why the carbonates we sampled are almost certainly Miocene (and in most cases late Miocene) in age.

*Line 50 - References to the overall and specific geology composition of the Ogalla are very dated, which isn't necessarily a problem except that it not consulting more recent studies likely led to the additional lithostratigraphic concerns listed below. The authors should carefully read the more recent papers specifically addressing the lithofacies and calcrete stratigraphy of the Ogallala such as:*
*Joeckel, R.M., Wooden Jr, S.R., Korus, J.T., and Garbisch, J.O., 2014, Architecture, heterogeneity, and origin of late Miocene fluvial deposits hosting the most important aquifer in the Great Plains, USA: Sedimentary Geology, v. 311, p. 75-95.*
*Smith, J.J., and Platt, B.F., 2023, Reconstructing late Miocene depositional environments in the central High Plains, USA: Lithofacies and architectural elements of the Ogallala Formation: Sedimentary Geology, v. 443, p. 106303.*
*Korus, J., and Joeckel, R.M., 2023, Telescopic Megafans on the High Plains, USA Were Signal Buffers in a Major Source-To-Sink System: The Sedimentary Record, v. 21.*

> We have now incorporated all of these references into the manuscript (Joeckel et al. 2014 was previously referenced).

*Line 52 - The concept of the Ogallala ''cap rock'' referring to a regionally persistent and ledge-forming terminal petrocalcic horizon is not supported by more recent stratigraphic studies (Swineford et al. 1958; Diffendal 1982; Gustavson and Winkler 1988; and Joeckel et al. 2014). Instead, these studies show that carbonate-cemented paleosols and petrocalcic horizons are present in numerous stratigraphic positions in Ogallala deposits. I understand that there is a very prominent calcrete at the contact between the Ogallala and the Blackwater Draw Formation in TX, but this should not be interpreted to represent a regionally persistent marker bed throughout the expanse of the Ogallala as was previously assumed (prior to the 2000s). See References above and Ludvigson et al. (2009), Review of the stratigraphy of the Ogallala Formation and revision of Neogene ("Tertiary") nomenclature in Kansas.*

> We have changed the wording here to indicate that the "cap-rock" is a regional distinctive

feature limited largely to the southern High Plains. We do not mean to imply anything regarding its genesis here; rather, we only wish to note that it is a regionally distinctive geomorphic feature (see also changes to lines 68-70).

*Line 68 - see earlier comment, while I agree Ogallala exposures are typically well indurated by carbonate, I would hesitate to refer to this characterstic as "its caprock", as its not a single bed.*

We have modified this sentence to denote that the many calcic-rich units help to create the characteristic escarpments of the High Plains.

*Figure 1- giving the circles and squares different colors might help to distinguish them a bit better.*

Great idea! We've modified the colors to help distinguish them.

*Line 175 – Some very interesting papers have recently been published on just this topic, see Korus, J., and Joeckel, R.M., 2023, Telescopic Megafans on the High Plains, USA Were Signal Buffers in a Major Source-To-Sink System: The Sedimentary Record, v. 21.*

We have now incorporated this reference throughout the manuscript.

*Line 182 – Additional publications with specific volcanic age data from Ogallala ash bed should be cited:*
*Swisher, C. C. III, 1992, 40Ar/39Ar dating and its application to the calibration of the North American land mammal ages [Ph.D. thesis]: Berkeley, University of California, 239 p.*
*Smith, J.J., Turner, E., Moller, A., Joeckel, R.M., and Otto, R.E., 2018, First U-Pb zircon ages for late Miocene Ashfall Konservat-Lagerstatte and Grove Lake ashes from eastern Great Plains, USA: Plos One, v. 13.*

Now added.

*Line 192 - This may be the case in some areas, but it is not a consistant feature. Calcretes are present in many Ogallala outcrops regardless of stratigraphic position as it is most like a result of exposure case hardening. See Joeckel et al. 2014 and Smith and Platt 2023 for more on modern interpretation of "cap rock".*

We have now modified this sentence to note that this is a feature only observed in the southern High Plains. We do not disagree that, north of Texas and New Mexico, caliches may result from case-hardening (though see arguments in our responses below about why the isotopic evidence suggests that this may not the case). Rather, in Texas and New Mexico another explanation for the caprock—which fits with the available field observations (Gustavson, 1996; Gustavson and Winkler, 1988)—is provided by Brock and Buck (2009), who posit that the Stage VI caliches that are typical of the southern High Plains result from extended landscape stability and continuous carbonate dissolution/precipitation (note though that Brock and Buck (2009) conduct their study on a different caprock in northwest Arizona).

*Line 205 – "on the assumption that the caprock formed simultaneously across the Great Plains." –*

*we know this is incorrect. See references above.*

> We have modified this sentence. While our original intention was simply to note that previous authors (not us) have relied upon such a method to provide temporal constraints, we realize the wording was confusing. We have now modified the sentence to note that certain Ogallala outcrops (particularly in New Mexico, where they are disconnected from the escarpment) have only ever been "dated" using geomorphoric or lithologic correlations across hundreds of kilometers (Frye et al., 1982).

*Line 211 – See Smith and Platt (2023) for more on unconformities and the thorny issue of Ogallala calcretes…*

*Line 226 - The authors need to provide more information on their samples and sampling methods. There are many carbonate morphologies in the Ogallala, and its becoming increasingly clear most are not coincident with paleosol formation. The pics in Fig. 2 helps, but is also concerning. Carbonate nodules and burrows may be authogenic, but I have some reservations about the pictured root casts and am very skeptical of the "cap rock" calcrete. We strongly suspect that many of these calcretes and calcrete morphologies are primarily carbonate precipitation due to case hardening of the exposed surface and not syndepositional. This is a vitally important issue because this may be the primary reason you are getting a consistently modern signal from your d18O; you may be sampling carbonates that precipitated essentially in response to recent exposure and under essentially modern conditions. I'm not stating that is the case, but its impossible for me to tell without being more specific in how and what you sampled.*

> We now include several sentences detailing the types of samples collected and our field and laboratory sampling methods. All sample types (for each sample) are listed in Table S1 (though this Table should be publicly available shortly, here is a link to access it prior to publication:
> https://datadryad.org/stash/share/sY5SXxVH-Hfp104CZ4m-grrcnOh-RAdwdbCNrF1Avcg).

> Because this comment questions perhaps the most critical assumption in our study (*i.e.*, that the sampled carbonates record late Miocene climate)—and, indeed, a critical assumption in nearly all paleoclimate studies that use stable isotopes—we also respond to it in detail here.

> First, while Smith and Platt (2023) and Joeckel et al. (2014) provide compelling evidence that some of the carbonates in some sections of the Ogallala are not syndepositional, the isotopic data presented herein does not necessarily support this interpretation. In each individual section that we present, though we sampled a wide-variety of carbonate types (ie, rhizoliths, nodules, burrows, matrix, and caliche/cap-rock), the $\delta^{18}O$ is nearly identical between sample types. This is perhaps best seen in Figure 4b (and also Table 1) in our manuscript, where we plot the $1\sigma$ for each section's $\delta^{18}O$. The $1\sigma$ values are very low (*i.e.*, <1 ‰), which indicates that in most of our sections, the variety of sampled carbonates have very similar $\delta^{18}O$ values. In only three sections is the $1\sigma > 1‰$. The section with the largest $1\sigma$ (WC; Wildcat Bluff Nature Park outside Amarillo, TX) has only 3 samples. The topmost sample (the carbonate-cemented ash, dated by Cepeda and Perkins (2006)) is almost certainly altered, with $\delta^{18}O$ and $\delta^{13}C$ values distinctly different than nearly all other samples in the

southern Great Plains (see Supplemental Table 1). The section with the second largest 1σ (ESP, from the Santa Fe Group in the Española basin) has a very wide $\delta^{18}O$ range, which has been the subject of further work by our research group (Bui et al., 2023; Spaur, 2022; Spaur et al., 2022) and is anyway outside the area of Ogallala deposition. This uniformity of $\delta^{18}O$ values in individual sections has been found by previous workers as well (Fox and Koch, 2004; Ludvigson et al., 2016). The fact that $\delta^{18}O$ is invariant in these sections suggests that all carbonates are forming from the same source waters. Thus, if rhizoliths, burrows, and nodules are original and formed syn-depositionally (as suggested by Joeckel et al. (2014)), then—from an isotopic perspective—the sampled caliches and caprock formed at the same time as the rhizoliths, burrows, and nodules.

Alternatively, climate could have been invariant (partly the hypothesis in our study) and all of these carbonates are simply recording modern meteoric water $\delta^{18}O$. However, here the $\delta^{13}C$ data strongly indicate that, for most of our sections, the carbonates that we sampled (though see note below regarding the sections in New Mexico) formed no later than the latest Miocene. In the Great Plains, there is a well-documented increase in carbonate $\delta^{13}C$ due to the spread of C4 grasslands after the Miocene (Fox and Koch, 2004, 2003). The appearance of C4 grasses leads to $\delta^{13}C$ values of approximately -2‰ (or even higher) by the Pleistocene. Nearly all of our Ogallala sections have mean $\delta^{13}C$ values < -6‰, indicating they formed in the late Miocene prior to the widespread dominance of C4 grasses. Indeed, some of the lowest mean $\delta^{13}C$ values are in the northern Great Plains. (We recognize that the $\delta^{13}C$ data were not available in the initial submission but they are now listed in Table 1 and publicly available via the Dryad link).

The only sections with $\delta^{13}C$ values > -6‰ occur in New Mexico. These sites have less precise age control than sites to the east in Texas and to the north. Most of these sites were originally studied by Frye et al. (1982), who correlated these sites to the Ogallala Formation based upon their geomorphic position and/or their lithology. We know of no studies (except the Masters thesis by Henry (2017)) that have followed up to constrain the age of deposition at these sites. At several of these sites, not only is the mean $\delta^{13}C$ > -6‰, but the $\delta^{13}C$ 1σ is relatively high, largely due to the fact that the caprock sometimes has a much higher $\delta^{13}C$ (in other cases, the caprock caliche has similar $\delta^{13}C$ values to the rest of the sampled carbonates). At these sites, then, the $\delta^{13}C$ may support the contention of Joeckel et al. (2014) and Smith and Platt (2023) that some of the sampled carbonates formed millions of years after deposition and/or that the caprock has a multi-genetic history (also found by Henry (2017)).

However, because we have no other independent age data for these sites (except for CP, studied by Henry (2017)), we are hesitant to exclude these data solely based on their $\delta^{13}C$ values and instead choose to include these data in our study. Further, there are also samples at many of these sites with low $\delta^{13}C$. Thus, we are hesitant to exclude these sections since, in many cases, these sites have carbonate samples that return $\delta^{13}C$ values indicative of formation during the late Miocene. Because there may have been landscape-scale variability in the abundance of C4 in the late Miocene (Lukens and Fox, 2022), it seems prudent to not exclude this data. We further note that excluding these data would not substantially alter the estimated mean $\delta^{18}O$ or modify the conclusions of this study.

Thus, we suggest that the largely invariant $\delta^{18}O$ in any given section indicates that all of the carbonates in any given section formed from the same meteoric water (or that climate has been relatively invariant since the late Miocene) and that the low $\delta^{13}C$ (low relative to modern soil carbonate $\delta^{13}C$) indicates that, in most of these sections, the carbonates must have formed prior to the spread of C4 grasses in the Pliocene. An interesting follow-up study would be to try and reconcile both the field observations of Joeckel et al. (2014) and Smith and Platt (2023) with the isotopic evidence for Miocene formation of carbonates from this study, particularly in the central and northern Great Plains. Additional work is also necessary to provide independent age estimates for many of the sites in New Mexico identified as the Ogallala Formation by Frye et al. (1982).

We have now included this reasoning in the manuscript in lines 504-554.

*Line 317 – "this year".... What year? 2016? Or an average of 1980-2016?*

We have modified the text to note that these plots encompass all months of the year (ie, January through December), averaged over the timeframe of the HYSPLIT climate model data (*i.e.*, 1980-2016).

*Line 421 – "also imprecision in the chronologies of the sections we sampled"... Not just the sections, but the sampled material itself, as in assuming the carbonate is in some respect syndepositional with the host sediments. I would be curious to see inter-area sampling differences. For example what is the variance in d18O between the 19 samples from the BV location? Are there patterns with respect to sample type (nodules that appear pedogenic vs calcrete vs rhizoliths)?*

We address the point about chronology imprecision in the point above, but do note that, while our samples record late Miocene formation (ie, prior to the spread of C4 grasslands), within this epoch, samples may not have necessarily formed syn-depositionally.

The 1σ for the BV location is 0.45 ‰ and the full range is 1.8 ‰ (reported in Table 1). As mentioned above, there is very little variance in the isotope data from any given section. It is difficult to compare $\delta^{18}O$ of sample types across sections due to the fact that $\delta^{18}O$ varies by more than 10‰ from our southernmost to our northernmost sites. However, within each section, there is no pattern in $\delta^{18}O$ with respect to sample type.

*Again, I apologize for not completing my review. I have few concerns about the results of the geochemical analyses. The methodology and output is well communicated, and I would not be surprised by their findings in the slightest; in fact they align very well with my most recent publication using paleosols and trace fossils to interpret climate conditions (Platt, B.F., and Smith, J.J., 2023. Late Miocene paleoecology and paleoclimate in the central High Plains of North America reconstructed from paleopedological, ichnological, and stable isotope analyses of the Ogallala Formation in western Kansas, USA. Evolving Earth, https://doi.org/10.1016/j.eve.2023.100019.) But frankly, I am extremely skeptical of their interpretations and conclusions due to the lack of communicating exactly what was sampled, how or why they suspect the sampled carbonate is ancient, and the authors out dated understanding of the regional geology. I was not able to complete*

*my review, so I will not make a final recommendation.*

We hope our revisions have helped to address these concerns, and we also incorporated the Platt and Smith (2023) reference into the manuscript. We have modified the manuscript to incorporate newer sedimentological interpretations and have revised our descriptions of the litho-stratigraphy. Regarding the stable isotope analysis, our data do not suggest that there are carbonates within most of these sections that formed at a substantially different time than the other carbonates in these sections. This conclusion arises due to the small variance in $\delta^{18}O$ in each section (suggesting all carbonates are recording the same waters) and that the $\delta^{13}C$ is low and clearly formed prior to the well-documented spread of C4 grasses in the Great Plains (though note the additional independent chronology work that is needed in New Mexico). These observations indicate that the carbonates sampled and analyzed in our study do indeed record late Miocene climate.

**References cited in review**:

Brock, A.L., Buck, B.J., 2009. Polygenetic development of the Mormon Mesa, NV petrocalcic horizons: Geomorphic and paleoenvironmental interpretations. Catena 77, 65–75. https://doi.org/10.1016/j.catena.2008.12.008

Bui, T., Spaur, S., Koning, D.J., Rugenstein, J.K.C., 2023. Terrestrial Stable Isotope Record of Miocene Monsoon and Mid-latitude Westerly Dynamics in the Southwest US, in: AGU Fall Meeting Abstracts. American Geophysical Union, San Francisco, CA.

Cepeda, J.C., Perkins, M.E., 2006. A 10 million year old ash deposit in the Ogallala Formation of the Texas Panhandle. Texas J. Sci. 58, 3–12.

Fox, D., Koch, P.L., 2004. Carbon and oxygen isotopic variability in Neogene paleosol carbonates: constraints on the evolution of the C4-grasslands of the Great Plains, USA. Palaeogeogr. Palaeoclimatol. Palaeoecol. 207, 305–329. https://doi.org/10.1016/j.palaeo.2003.09.030

Fox, D.L., Koch, P.L., 2003. Tertiary history of C4 biomass in the Great Plains, USA. Geology 31, 809–812. https://doi.org/10.1130/G19580.1

Frye, J.C., Leonard, A.B., Glass, H.D., 1982. Western extent of Ogallala Formation in New Mexico. New Mex. Bur. Mines Miner. Resour. Circ. 175, 41.

Gustavson, T.C., 1996. Fluvial and Eolian Depositional Systems, Paleosols, and Paleoclimate of the Upper Cenozoic Ogallala and Blackwater Draw Formations, Southern High Plains, Texas and New Mexico, Report of Investigations. Austin, TX.

Gustavson, T.C., Winkler, D.A., 1988. Depositional facies of the Miocene-Pliocene Ogallala Formation, northwestern Texas and eastern New Mexico. Geology 16, 203–206. https://doi.org/10.1130/0091-7613(1988)016<0203

Henry, K.D., 2017. 40Ar/39Ar Detrital Sanidine Dating of the Ogallala Formation, Llano Estacado, Southeastern New Mexico and West Texas. New Mexico Tech.

Joeckel, R.M., Wooden, S.R., Korus, J.T., Garbisch, J.O., 2014. Architecture, heterogeneity, and origin of late miocene fluvial deposits hosting the most important aquifer in the great plains, USA. Sediment. Geol. 311, 75–95. https://doi.org/10.1016/j.sedgeo.2014.07.002

Ludvigson, G.A., Mandel, R., MacFarlane, A., Smith, J.J., 2016. Capturing the Record of Neogene Climate Change from Strata of the High Plains Aquifer in Kansas: Research Activities and Findings.

Lukens, W.E., Fox, D.L., 2022. Paleovegetation and paleo-pedogenic properties from the upper Miocene Coffee Ranch fossil site in the North American Great Plains. Palaeogeogr.

Palaeoclimatol. Palaeoecol. 586, 110760. https://doi.org/10.1016/j.palaeo.2021.110760

Platt, B.F., Smith, J.J., 2023. Late Miocene paleoecology and paleoclimate in the central High Plains of North America reconstructed from paleopedological, ichnological, and stable isotope analyses of the Ogallala Formation in western Kansas, USA. Evol. Earth 1, 100019. https://doi.org/10.1016/j.eve.2023.100019

Quade, J., Roe, L.J., 1999. The stable-isotope composition of early ground-water cements from sandstone in paleoecological reconstruction. J. Sediment. Res. 69, 667–674.

Smith, J.J., Platt, B.F., 2023. Reconstructing late Miocene depositional environments in the central High Plains, USA: Lithofacies and architectural elements of the Ogallala Formation. Sediment. Geol. 443, 106303. https://doi.org/10.1016/j.sedgeo.2022.106303

Spaur, S., 2022. The hydroclimate and environmental response to warming in the southwestern US: A study across the mid-Miocene Climate Optimum. Colorado State University.

Spaur, S., Rugenstein, J.K.C., Koning, D.J., Heizler, M., Aby, S.B., 2022. Miocene climate dynamics of the Rio Grande Rift region, in: Geological Society of America Abstracts with Programs. Geological Society of America, Denver, Colorado.

---

## Author Comment (AC4)

Jeremy K. Caves Rugenstein
Assistant Professor
Department of Geosciences
1482 Campus Delivery
Fort Collins, Colorado 80523-1482
(505) 910-7918; FAX: (970) 491-6307
www.warnercnr.colostate.edu/Geosciences-Home

January 29, 2024

Dear Dr. Reyes,

Thank you for handling our manuscript. We have responded to the reviewer comments, which have substantially improved our interpretation and also our description of the Ogallala Formation more generally. Though we go into detail in response to each of the reviewers' comments, we summarize here three major points raised by the reviewers and our responses:

1. We have substantially revised our description of the Ogallala Formation in Section 3 (we note that we use the terminology "Formation" throughout the manuscript) based upon the feedback from the reviewers. We have incorporated *(1)* more accurate descriptions of the calcic units found throughout the Ogallala, *(2)* additional references, and *(3)* highlighted the regional variations in the Ogallala, particularly between the southern and northern High Plains.

2. We now report (in Table 1 and Table 3) the $\delta^{13}C$ of our data and of the compiled carbonate data. We now also provide a statistical overview of this data in Section 5.2. Table S1 also contains all of the $\delta^{13}C$ data. If the doi remains unavailable until publication, please use the following link to access Table S1: https://datadryad.org/stash/share/sY5SXxVH-Hfp104CZ4m-grrcnOh-RAdwdbCNrF1Avcg

3. We now address, in Section 6.2.3, why our stable isotope data suggest that these carbonates formed in the late Miocene. In short, there is very little variation in the $\delta^{18}O$ values of our carbonates and the uniformity of $\delta^{18}O$ values spans different types of samples, suggesting that all of the carbonates formed from the same source waters. Additionally, in nearly all of our sections (except those in eastern New Mexico, see response to reviewers for further details), the $\delta^{13}C$ values are low and much lower than published Pliocene and Pleistocene carbonate $\delta^{13}C$ values that record the spread of C4 grasses in the Plio-Pleistocene. Consequently, we suggest that nearly all of our samples formed in the late Miocene and record late Miocene environmental conditions.

We look forward to further comments and feedback on our manuscript and again thank you and the reviewers for your thoughtful and helpful comments on our study.

Sincerely and on behalf of all authors,

Jeremy K. Caves Rugenstein

---

## Author Response (AR2)

Jeremy K. Caves Rugenstein
Assistant Professor
Department of Geosciences
1482 Campus Delivery
Fort Collins, Colorado 80523-1482
(505) 910-7918; FAX: (970) 491-6307
www.warnercnr.colostate.edu/Geosciences-Home

March 17, 2024

Dear Dr. Reyes,

We have now fixed the minor points raised by Dr. Fox (see line-by-line responses below). Thank you again for handling our manuscript, and please let us know if any additional files are necessary for acceptance of the manuscript.

Sincerely and on behalf of all authors,

Jeremy K. Caves Rugenstein

**Responses to Minor Comments by Dr. Fox**
*Line 87: change "paleoclimate data indicates" to "paleoclimate data indicate" (data is plural).*

> Fixed.

*Line 165: change "contains low d18O" to "has low d18O [values]" (d18O is a value, not an object)*

> Fixed.

*Line 168: "encoded in the d18O data"...Which data do you mean? This is a little inexact.*

> Fixed. We have now clarified this by writing, "The aridity gradient, which depends on the relative contributions of westerly vs GPLLJ moisture, is therefore encoded in the value of $\delta^{18}O_p$ across the Great Plains, which can generally be understood as the precipitation-weighted average of the end-member sources."

*Line 200: change "Plio-Pleistocene the Blackwater" to "Plio-Pleistocene Blackwater"*

> Fixed.

*Line 584: change "also do to" to "also due to"*

> Fixed.